# Molecular mechanism of ultrafast transport by plasma membrane Ca$^{2+}$-ATPases

Deivanayagabarathy Vinayagam[1,5], Oleg Sitsel[1,4,5], Uwe Schulte[2,5], Cristina E. Constantin[2], Wout Oosterheert[1], Daniel Prumbaum[1], Gerd Zolles[2], Bernd Fakler[2,3 ✉] & Stefan Raunser[1 ✉]

Tight control of intracellular Ca$^{2+}$ levels is fundamental as they are used to control numerous signal transduction pathways[1]. Plasma membrane Ca$^{2+}$-ATPases (PMCAs) have a crucial role in this process by extruding Ca$^{2+}$ against a steep concentration gradient from the cytosol to the extracellular space[2]. Although new details of PMCA biology are constantly being uncovered, the structural basis of the most distinguishing features of these pumps, namely, transport rates in the kilohertz range and regulation of activity by the plasma membrane phospholipid PtdIns(4,5)P$_2$, has so far remained elusive. Here we present the structures of mouse PMCA2 in the presence and absence of its accessory subunit neuroplastin in eight different stages of its transport cycle. Combined with whole-cell recordings that accurately track PMCA-mediated Ca$^{2+}$ extrusion in intact cells, these structures enable us to establish the first comprehensive transport model for a PMCA, reveal the role of disease-causing mutations and uncover the structural underpinnings of regulatory PMCA–phospholipid interaction. The transport cycle-dependent dynamics of PtdIns(4,5)P$_2$ are fundamental for its role as a 'latch' promoting the fast release of Ca$^{2+}$ and opening a passageway for counter-ions. These actions are required for maintaining the ultra-fast transport cycle. Moreover, we identify the PtdIns(4,5)P$_2$-binding site as an unanticipated target for drug-mediated manipulation of intracellular Ca$^{2+}$ levels. Our work provides detailed structural insights into the uniquely fast operation of native PMCA-type Ca$^{2+}$ pumps and its control by membrane lipids and drugs.

P-type ATPases are a family of membrane proteins that derive energy from ATP hydrolysis to power the transport of various ions or lipids across membranes against their concentration gradients[2]. Because of their essential function, these proteins are found in nearly all life-forms and include well-known members such as the Na$^+$/K$^+$-ATPase, the H$^+$/K$^+$-ATPase and the sarcoplasmic/endoplasmic reticulum Ca$^{2+}$-ATPase (SERCA). The importance of P-type ATPases is further highlighted by their involvement in many human diseases[3–9] and their targeting by drugs during therapeutic treatment of, for example, congestive heart failure[10] or dyspepsia[11]. The structural core of P-type ATPases, commonly augmented by accessory domains and subunits, is composed of an ion-translocating transmembrane (M) domain linked to the cytoplasmic actuator (A), phosphorylation (P) and nucleotide-binding (N) domains. The ATP hydrolysis and autodephosphorylation-driven interplay between the three cytoplasmic domains trigger conformational changes in the M-domain that lead to cytoplasmic ion binding, translocation, release and often counter-ion transport in the opposite direction[12]. This intricate transport scheme known as the Post–Albers cycle[13] occurs through phosphorylated intermediate states (hence the name P-type), and has historically been best studied in context of the Ca$^{2+}$-ATPase SERCA1a[14], which, along with secretory pathway Ca$^{2+}$-ATPases (SPCAs)[15],

belongs to the P-type ATPase subgroup P2A[2]. A similar understanding of the transport cycle for the distinct P2B subgroup of PMCAs is however lacking despite the recently reported structure of PMCA1 (ref. 16).

Unlike P2A ATPases, which sequester Ca$^{2+}$ in limited-capacity intracellular stores, PMCAs extrude Ca$^{2+}$ ions from the intracellular milieu to the extracellular space, often in concert with Na$^+$/Ca$^{2+}$-exchanger (NCX) proteins[17]. This extrusion is crucial for proper operation of Ca$^{2+}$ signalling, which constitutes a transient increase in intracellular Ca$^{2+}$ concentration that controls innumerable cellular processes in a precisely timed manner[17]. The role of PMCA in this context is the effective termination of Ca$^{2+}$ signals by ultrafast Ca$^{2+}$ extrusion[18]. This is particularly prominent in the brain, where signal transmission occurs at high frequencies of up to 1 kHz. PMCAs are encoded by four different genes, PMCAs 1–4, that are differentially expressed in virtually all cell and tissue types. PMCA1 and PMCA4 show ubiquitous expression, whereas PMCA2 and PMCA3 demonstrate a more restricted expression pattern, being predominant in the central nervous system. Native PMCAs form heterodimeric assemblies with the small transmembrane proteins neuroplastin (NPTN) and basigin (BASI)[19]. The only currently available PMCA1 structure contains an averaged NPTN–BASI density, which precluded detailed structural analyses[16].

[1]Department of Structural Biochemistry, Max Planck Institute of Molecular Physiology, Dortmund, Germany. [2]Institute of Physiology, Faculty of Medicine, University of Freiburg, Freiburg, Germany. [3]Signalling Research Centres BIOSS and CIBSS, Freiburg, Germany. [4]Present address: Marine Structural Biology Unit, Okinawa Institute of Science and Technology Graduate University, Okinawa, Japan. [5]These authors contributed equally: Deivanayagabarathy Vinayagam, Oleg Sitsel, Uwe Schulte. ✉e-mail: bernd.fakler@physiologie.uni-freiburg.de; stefan.raunser@mpi-dortmund.mpg.de

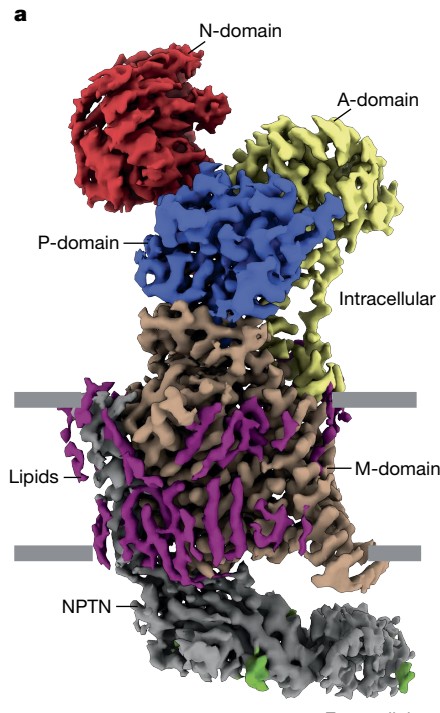

**Fig. 1 | Cryo-EM structures of PMCA2–NPTN in different states of the Post–Albers cycle. a**, Local-resolution filtered, sharpened cryo-EM density map of the PMCA2–NPTN complex in the E2.P$_i$ state. The N-domain, A-domain, P-domain and M-domain of PMCA2 are in red, yellow, blue and brown, respectively. NPTN is shown in grey. Densities corresponding to lipids and glycans are in purple and green, respectively. The cell membrane is indicated by grey bars. **b**, PMCA2–NPTN structures in six different states of the Post–Albers cycle. PtdIns(4,5)P$_2$ is shown in magenta. See also Supplementary Video 1.

The distinguishing features of PMCAs over other Ca$^{2+}$-ATPases are their regulation by phospholipids, most prominently phosphatidylinositol 4,5-bisphosphate (PtdIns(4,5)P$_2$)[20,21], and their transport rates of more than 5,000 cycles per second[18] under cellular conditions. These ultrafast rates are different from other P-type ATPases including the P2A Ca$^{2+}$ pumps, which operate in the range of tens to hundreds of cycles per second, and are more comparable with those of secondary transporters[22–24].

To understand the distinguishing features and the operation of these classical Ca$^{2+}$ pumps of the plasma membrane at the molecular level, we determined an ensemble of PMCA2 structures trapped in different stages of the Post–Albers cycle by cryo-electron microscopy (cryo-EM), in both the presence and the absence of its accessory subunit NPTN. Combined with whole-cell patch-clamp recordings that monitor activity of PMCA2–NPTN complexes in intact cells, we are able to describe the complete PMCA transport cycle and explain how these proteins interact with PtdIns(4,5)P$_2$. Our study further explains how this phospholipid, together with ATP hydrolysis-driven structural rearrangements, enables the high speed of these vital primary transporters under normal conditions and how it is impaired by loss-of-function mutations in PMCA2 that lead to autosomal dominant deafness[25,26].

## Overall architecture and dynamics

We first expressed mouse PMCA2 in *NPTN/BSG*-knockout cells (*BSG* encodes BASI) to obtain a homogeneous population of target protein for structural analyses, enabling us to isolate PMCA2 both alone and exclusively complexed with NPTN when the latter was co-expressed. Modulating the composition of the final elution buffer produced preparations of PMCA2 or PMCA2–NPTN trapped in various intermediate states of the transport cycle, which were then plunge frozen and their structures determined by single-particle cryo-EM

(Extended Data Table 1, Supplementary Table 1 and Supplementary Figs. 1–5).

The eight resulting structures of PMCA2 had an overall resolution ranging from 2.8 Å to 3.6 Å. Although the densities for the cytoplasmic domains, in particular the A-domain, are weaker in some of the reconstructions, indicating flexibility, the M-domain within the PMCA2 reconstructions is resolved up to 2.5 Å. This enabled the accurate modelling of side chains, the phospholipid PtdIns(4,5)P$_2$ and ordered water molecules. The structures revealed an overall classical P-type ATPase architecture with ten transmembrane helices characteristic of the P2 subgroup (Fig. 1a,b). The A-domain is located between helices M1 and M2 (numbered in ascending order starting from the N terminus), the P-domain is positioned between M4 and M5, and the N-domain is inserted into the P-domain. Most known disease-causing point mutations of PMCAs cluster in these three cytoplasmic domains (Extended Data Fig. 1a).

With regard to the Post–Albers cycle, the structures that we obtained correspond to the inward-open E1, nucleotide-bound E1-ATP, ion-bound E1-Ca, ion- plus nucleotide-bound E1-Ca-ATP, outward-open phosphorylated E2P and outward-open dephosphorylating E2.P$_i$ states[27] (Fig. 1b and Extended Data Fig. 1b). Comparison of the different structures allowed us to follow crucial conformational changes during the catalytic cycle (Fig. 2 and Supplementary Video 1). Binding of the nucleotide to the interface of the PMCA2 N-domain, P-domain and A-domain induces conformational changes that prime the ion-binding site (Fig. 2a). Independent binding of Ca$^{2+}$ results in the closure of the intracellular gate (Fig. 2b). Upon ATP hydrolysis, PMCA2 undergoes major structural rearrangements in the cytoplasmic domains (Fig. 2c and Supplementary Video 2), causing the opening of the M-domain to the extracellular space (Fig. 2d and Supplementary Video 3). In turn, this triggers the release of ADP and Ca$^{2+}$, and the binding of counter-protons, finally leading to the E2P state. The transition to the dephosphorylation

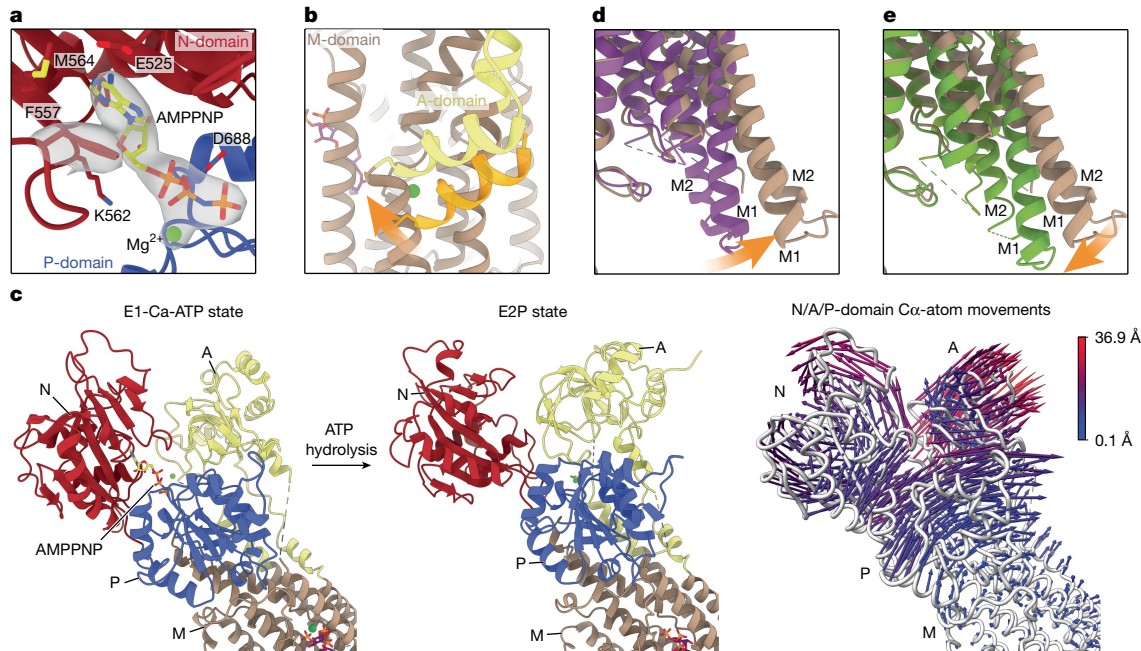

**Fig. 2 | Dynamic actions observed in the PMCA2–NPTN structures.**
**a**, Binding of nucleotide to the interface of the three cytoplasmic domains after transition from the E1 to the E1-ATP state. **b**, Closure of the intracellular ion entry pathway (open in the E1 state (orange)) upon ion binding in the E1-Ca state (yellow). **c**, Conformational changes of the three cytosolic domains between the E1-Ca-ATP and E2P states. Movement of Cα atoms between the two states is shown by arrows (right). See also Supplementary Video 2. **d**, Opening of the extracellular Ca²⁺-release pathway after transition from the E1-Ca-ATP (purple) to the E2P (brown) state. See also Supplementary Video 3. **e**, Closure of the extracellular Ca²⁺-release pathway after transition from the E2P (brown) to the E2.Pᵢ (green) state. See also Supplementary Video 3. In panels **c**–**e**, dashed lines indicate disordered amino acid regions.

intermediate E2.Pᵢ exhibits relatively minor movements, mainly pertaining to the closure of the ion-release pathway of the M-domain (Fig. 2e and Supplementary Video 3). Finally, the transition between the E2.Pᵢ and E1 state upon dephosphorylation rearranges the cytoplasmic domains and causes the M-domain to return into its cytoplasm-open state, releasing the counter-protons and closing the Post–Albers cycle (Supplementary Video 1).

The nearly indistinguishable E1-Ca and E1-ATP states of PMCA2 represent two different pathways that the ATPase can take towards the E1-Ca-ATP state depending on local ATP availability[28]. In the PMCA2 E1-ATP structure, we observed strong, unambiguous density for the bound nucleotide in the absence of Ca²⁺. By contrast, in crystal structures of SERCA1a in the E1-ATP state[29], the density for the added nucleotide is extremely weak (Extended Data Fig. 2a,b). This suggests that, when utilizing the E1-ATP pathway, PMCAs can stably associate with the nucleotide already before Ca²⁺ binds, whereas SERCAs cannot.

Regardless of which pathway may be used to reach the ion-bound state, the 'base lid' of the A-domain controlling access to the intracellular ion entry pathway closes immediately upon Ca²⁺ binding (Fig. 2b). This promptly prevents Ca²⁺ from dissociation, and is in stark contrast to P2A-ATPases where this closure occurs only after binding of the nucleotide. Another notable difference between these Ca²⁺-pump subtypes is that Ca²⁺ binding to the E1 state of PMCA2 leads to only minor rearrangements of the N-domain and P-domain with rotations and shifts, respectively, of 8.3° and 2.2 Å and 4.2° and −0.1 Å. In P2A-ATPases exemplified by SERCA1a, the corresponding changes are 66.5° and 9.0 Å and 12.0° −0.1 Å (Supplementary Table 2). Consequently, the soluble domains of PMCA2 need to move much less in transition from either the E1-Ca or E1-ATP state to the E1-Ca-ATP state. In addition, the cytoplasmic domains of PMCA2 interact with each other using markedly smaller surface areas than other Ca²⁺-ATPases (Supplementary Table 3). This is true for all determined states of the PMCA2 transport cycle and applies to both the overall interaction area of the cytoplasmic domains, and to the interactions between individual domains. Together, the resulting decrease in energy required for the conformational changes during the catalytic cycle is probably an essential basis for the uniquely fast operation of PMCA-type Ca²⁺ pumps.

## Interaction with neuroplastin

NPTN was clearly resolved in all NPTN–PMCA2 structures, including the immunoglobulin-like domains Ig2 and Ig3. The most N-terminal Ig1 domain, however, was highly flexible and consequently less resolved. We therefore determined its structure by X-ray crystallography and fitted the resulting 2 Å structure (Extended Data Table 2) into the density of the E1-Ca state (Fig. 1b and Supplementary Fig. 1). The resulting combined atomic model of NPTN represents, to our knowledge, the first complete structure of NPTN (Fig. 3a). We identified densities for six N-linked glycans in Ig2 and Ig3 (N170, N196, N228, N283, N295 and N316) that do not directly interact with PMCA2.

The accessory NPTN subunit associates with PMCA2 via two interfaces (Fig. 3a–c). The most extensive contact takes place near the membrane plane where the transmembrane helix of NPTN binds to helix M10 and the cytoplasm-proximal loop between helices M8 and M9 of PMCA2 via numerous hydrophobic contacts (Fig. 3b). At the second interface, four residues of the NPTN Ig3 domain interact with the extracellular M7–M8 loop of the pump (Fig. 3c). These interactions occur via polar contacts and hydrogen bonds, and thus partially tether the extracellular regions of NPTN to that of PMCA2. The Ig2 and Ig1 domains of NPTN do not interact directly with PMCA2, but Ig2 appears rigidly tethered to Ig3 by means of two hydrogen bonds (Fig. 3d). It is noteworthy that the PMCA–NPTN interfaces are highly conserved among PMCA subtypes (Extended Data Fig. 3a), suggesting that there is no subtype selectivity for NPTN. Our structures combined with sequence conservation analysis also provide a straightforward explanation for the finding that embigin, a member of the basigin group of Ig superfamily proteins along with NPTN and basigin[30], does not bind to native PMCAs[19]: embigin lacks several residues, which we identified

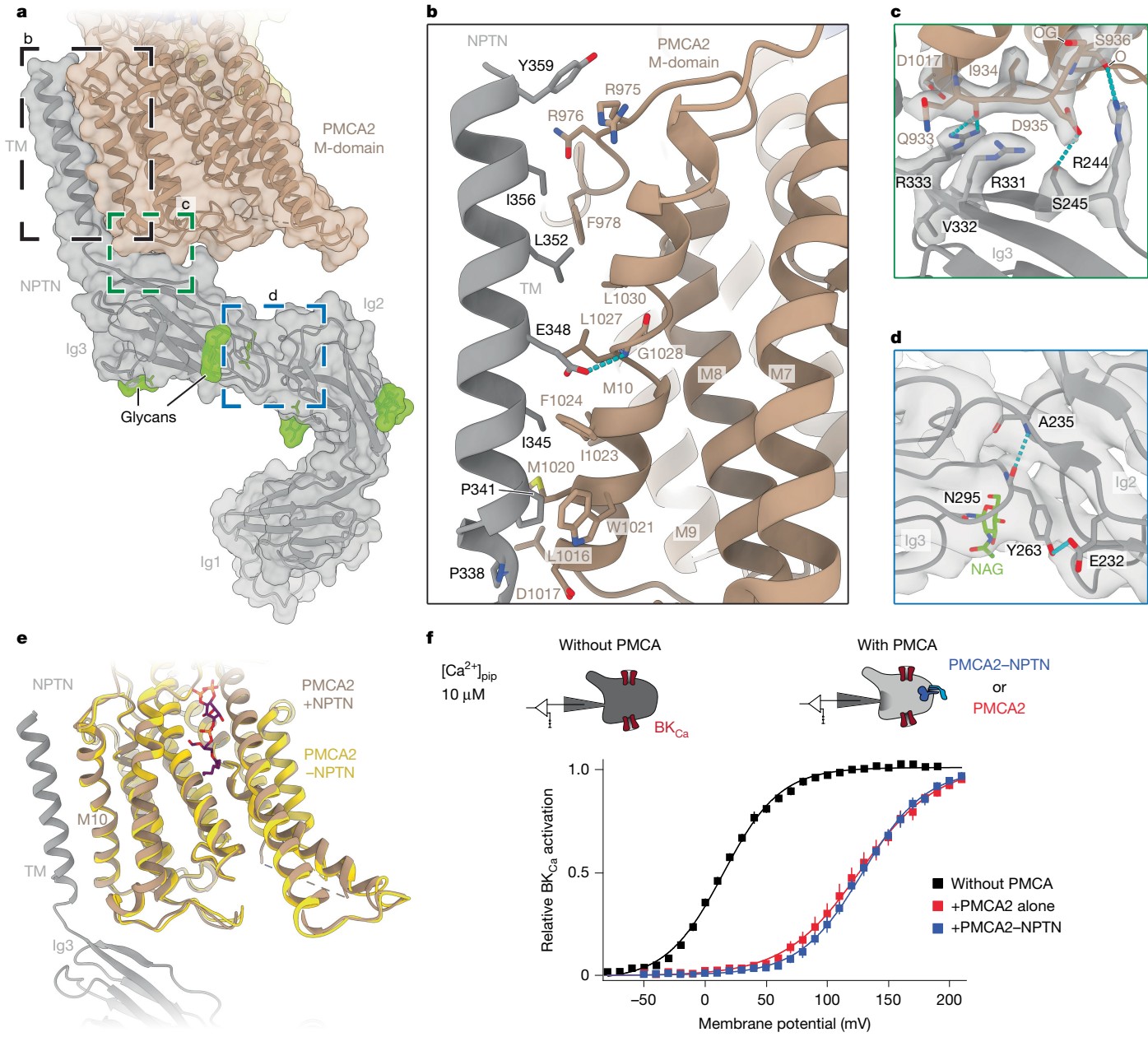

**Fig. 3 | Interactions of PMCA2–NPTN. a**, Transmembrane (TM) and extracellular interaction interfaces of PMCA2 and NPTN in the E2.P$_i$ state structure. Both proteins are shown as a semi-transparent surface and cartoon. The Ig1 domain of NPTN was resolved by X-ray crystallography and docked into the cryo-EM density. The colours are as in Fig. 1. **b,c**, Zoomed-in view of the transmembrane interface (**b**) and the extracellular interface (**c**) between PMCA2 and NPTN. **d**, Zoomed-in view of the hydrogen-bond-mediated interlock between Ig2 and Ig3 of NPTN. NAG, *N*-acetyl-D-glucosamine. In panels **b**–**d**, hydrogen bonds are shown as cyan dashed lines. Protein models are shown with their corresponding cryo-EM density (**c**,**d**). **e**, Overlay between structures of PMCA in the E2P state with (brown) and without (gold) NPTN. Cα root-mean-square deviation = 0.96 Å for E1-Ca-ATP ± NPTN and 0.82 Å for E2P ± NPTN. **f**, Ca$^{2+}$ transport of PMCA2 is independent of NPTN. Steady-state activation curves of BK$_{Ca}$ channels recorded in whole-cell mode with 10 µM free Ca$^{2+}$ in the patch pipette (pip; top) from tsA cells (tsA *NPTN/BSG$^{-/-}$*) in the absence (squares and line in black) and presence of PMCA2 or PMCA2–NPTN complexes (squares and line in red and blue, respectively; bottom). The lines are a result of a Boltzmann function fitted to the data, pump-mediated shift of the activation curve indicating a reduction of [Ca$^{2+}$]$_i$ from pipette-delivered 10 µM to values below 0.1 µM (see also Extended Data Fig. 4). Data represent mean ± s.e.m.

as necessary for the PMCA2–NPTN interaction outside the membrane (Extended Data Fig. 3b).

The interactions between NPTN and PMCA2 do not notably change throughout the Post–Albers cycle (Extended Data Fig. 2b and Supplementary Video 1), indicating that NPTN is not actively involved in any of the crucial conformational changes during the transport cycle, nor does it regulate PMCA2 activity in a phospholamban-like and

sarcolipin-like manner as seen with SERCA Ca$^{2+}$-ATPases[31]. Moreover, solitary PMCA2 structures obtained from an NPTN-free environment in both the ion-bound E1-Ca-ATP state and the ion-free E2P state were indistinguishable from PMCA2 co-assembled with NPTN (Fig. 3e).

This finding is in contrast to previous work reporting inactivation of human PMCA1 (hPMCA1) by removal of the accessory subunits with the detergent DTAC[16]. Therefore, we directly probed the effect of NPTN

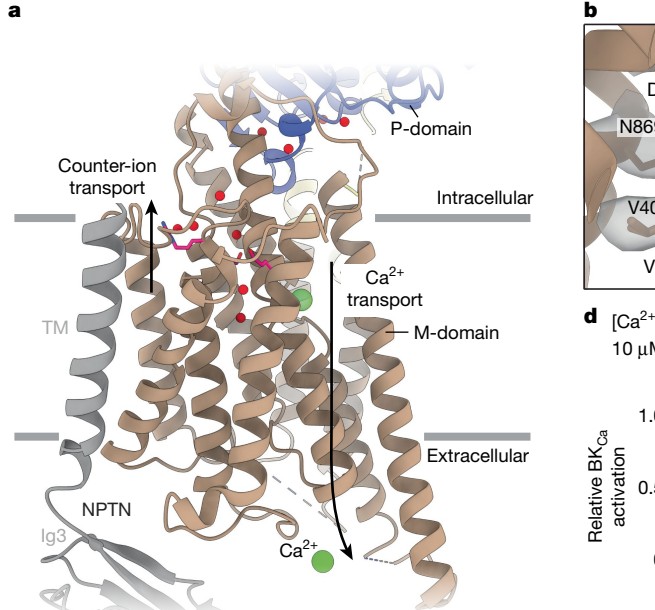

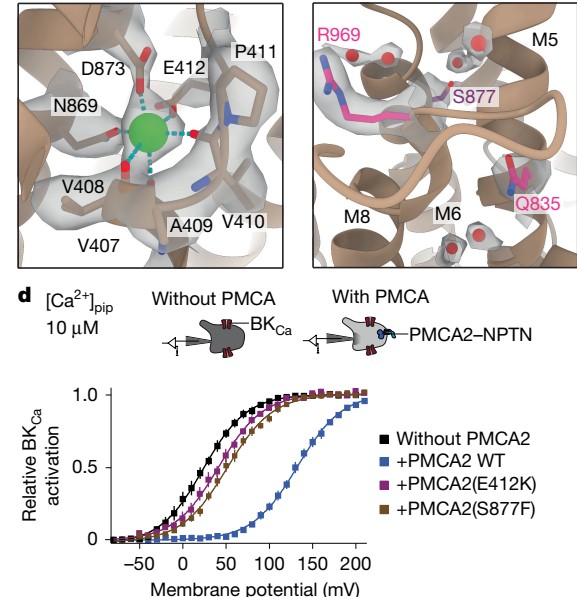

**Fig. 4 | Ion binding and counter-ion release in PMCA2. a**, Structure of the transmembrane region of PMCA2 in the E2.P$_i$ state. Directions of Ca$^{2+}$ transport and counter-ion transport are indicated with arrows. Water molecules are shown as red spheres. Although Ca$^{2+}$ was not present in the E2.P$_i$ state reconstruction, it is depicted at two positions in the model as a green sphere to emphasize the transport direction. **b**, Protein model and corresponding cryo-EM density of the ion-binding pocket of PMCA2 in the E1-Ca-ATP state. Protein–Ca$^{2+}$ interactions are shown as cyan dashed lines. **c**, Zoom of the putative counter-ion pathway in PMCA2. The pathway is lined by water molecules in the E2.P$_i$ structure and is a hotspot for disease-causing

mutations identified in PMCA1 (Q835R and R969Q in magenta) and PMCA2 (S877F in purple). Cryo-EM density is shown for the water molecules and respective side chains. **d**, PMCA2-mediated Ca$^{2+}$ transport is severely impaired by disease-causing mutations in the Ca$^{2+}$-binding site (E412K) and the counter-ion pathway (S877F). BK$_{Ca}$ activation curves recorded in whole-cell mode with 10 μM free Ca$^{2+}$ in patch pipette (top) as in Fig. 3f from CHO cells co-expressing PMCA–NPTN pumps harbouring either PMCA2 wild type (WT) or the indicated PMCA2 mutants. Note the largely reduced right shift in activation by the PMCA mutations compared with WT. Data are mean ± s.e.m. of 6–9 cells.

on PMCA-mediated transport by functional recordings from *NPTN/BSG*-knockout cells where PMCA2 was (over)expressed either alone or together with NPTN. Using Ca$^{2+}$-activated BK (BK$_{Ca}$) channels as native reporters of intracellular Ca$^{2+}$ concentration ([Ca$^{2+}$]$_i$)[19] (Extended Data Fig. 4a and Methods), we found that the pump-mediated shift in channel activation, a direct reflection of the resulting decrease in [Ca$^{2+}$]$_i$, was identical for both PMCA2 and PMCA2–NPTN complexes (Fig. 3f and Extended Data Fig. 4a,b). In line with our structural findings, these results indicate that NPTN does not directly impact the transport activity of PMCA2. Rather, the main cellular functions of NPTN appears to be the positioning and stabilization of the pump at the cell surface probably supported by the extensive N-glycosylation and promotion of robust trafficking of PMCAs from the endoplasmic reticulum to the plasma membrane[19]. Such trafficking does not occur with P2A-ATPases, which remain in intracellular membrane compartments, probably due to co-assembly with multiple partner proteins that secure their residence in the endoplasmic reticulum (Extended Data Fig. 4c,d).

## Ca$^{2+}$ binding and counter-ion transport

For a better understanding of the PMCA Ca$^{2+}$ transport mechanism, we analysed Ca$^{2+}$ binding in the E1-Ca and E1-Ca-ATP structures. We identified only one binding site formed by the backbones of residues V407, V408 and V410 and the side chains of residues E412, N869 and D873 (Fig. 4a,b). This binding site is equivalent to site II of the two Ca$^{2+}$-binding sites of SERCA (Extended Data Fig. 5a,b). The equivalent of SERCA site I is rendered non-functional in PMCA2 by two hydrophobic replacements (T799 by M872 and E771 by A844) similar to SPCA1 (refs. 15,32,33) and LMCA1 (ref. 34), explaining why Ca$^{2+}$ cannot bind at this position.

It is noteworthy that Ca$^{2+}$ was bound to PMCA2 despite the use of Ca$^{2+}$-free buffers for purification supplemented with 0.5 mM of the Ca$^{2+}$ chelators EGTA and EDTA (Methods). In fact, Ca$^{2+}$ was only released upon abundant addition of chelators (10 mM EGTA) or compounds forcing the pump into the phosphorylated E2 state, suggesting that PMCA2 has a high affinity for Ca$^{2+}$. These results are in stark contrast to SERCA, in which millimolar Ca$^{2+}$ concentrations were required to obtain structures of ion-bound states[35–37] despite the previously measured micromolar affinity of SERCA to Ca$^{2+}$ in sarcoplasmic reticulum vesicles[38]. The sole Ca$^{2+}$-binding site in PMCA2 should be disrupted by the E412K mutation that is linked to hereditary deafness[39] (Extended Data Fig. 1a). When recapitulated in heterologous expression experiments, this mutation indeed resulted in largely reduced transport activity in PMCA2–NPTN complexes, corroborating the results of our structural analysis (Fig. 4d).

By comparing the Ca$^{2+}$-binding site in the E1, E1-ATP and E1-Ca-ATP state structures, we found that nucleotide binding moves the residues of the ion-binding site into a configuration that is most suitable for the coordination of Ca$^{2+}$ (Extended Data Fig. 5a). This pre-positioning, although not required for substrate binding, ensures that Ca$^{2+}$ can bind most efficiently, thus streamlining the PMCA transport cycle. It is noteworthy that we did not observe a stabilizing Mg$^{2+}$ ion at the Ca$^{2+}$-binding site of PMCA2 in the Ca$^{2+}$-free E1 and E1-ATP states despite Mg$^{2+}$ being present at millimolar concentrations at above-neutral pH. This contrasts with SERCA1a (Extended Data Fig. 5b), the only other Ca$^{2+}$-ATPase for which structures of these states have been determined, where Mg$^{2+}$ stabilizes the corresponding ion-binding site[29,40]. We therefore conclude that PMCA2 has a much lower affinity for Mg$^{2+}$ than SERCA1a. The resulting absence of competition between these two ion species for the same binding site during the transport cycle

probably accelerates the transition of PMCA2 to the Ca$^{2+}$-bound states compared with SERCA1a.

Upon the release of Ca$^{2+}$ following rearrangement of the M-domain during the E1-to-E2 transition, PMCA2 is expected to bind counter-protons. In SERCA, these protons neutralize the negative charges in the empty ion-binding site[41] and are transported through a water-lined release pathway to the cytoplasm when the M-domain reverts back to the E1 state. In our structure of the E2.P$_i$ state (Fig. 4c), we detected several well-resolved water molecules in PMCA2 corresponding to the C-terminal counter-proton release pathway of SERCA. Although two water-lined release pathways[42] have been described for the two or three counter-protons of SERCA1a[43], we only found one pathway in PMCA2 (Fig. 4c and Extended Data Fig. 5c,d). A compact cluster of disease mutations (Q835R, S877F and R969Q) is located in close proximity to the water molecules of this release pathway (Fig. 4c and Extended Data Fig. 1a). The S877F exchange has been especially well characterized in Oblivion mice mutants where it causes deafness due to PMCA2 inactivation[44], whereas the other two exchanges were identified as disease mutants in humans[45]. When recapitulated in our cellular testing system, the S877F mutation profoundly reduced Ca$^{2+}$ pumping in PMCA2–NPTN complexes, similar to the E-to-K mutation in the Ca$^{2+}$-binding site, although the water molecules do not bind directly to S877 in our structure (Fig. 4c). This indicates that the S877F mutation probably alters the entire electrostatics of the counter-ion release pathway, its ability to bind to water molecules and to conduct protons. Together, these data suggest that PMCA2 uses the equivalent of the SERCA1a C-terminal pathway for counter-proton transport or release.

## The PtdIns(4,5)P$_2$-binding pocket

The unique activation of PMCAs by the plasma membrane phospholipid PtdIns(4,5)P$_2$ is an outstanding feature of these P-type ATPases documented mostly in vesicular preparations of purified PMCAs[46]. To probe PtdIns(4,5)P$_2$-mediated activation of PMCAs in vivo, we measured Ca$^{2+}$ transport of PMCA2–NPTN under intact cellular conditions before and after removal of the phospholipid from the membrane. This was done using the co-expressed voltage-activated PtdIns(4,5)P$_2$ phosphatase CiVSP that effectively depletes PtdIns(4,5)P$_2$ after activation of its enzymatic activity by membrane depolarization[47] (Methods). PtdIns(4,5)P$_2$ depletion largely reduced the PMCA-mediated right-shift of the BK$_{Ca}$ channel activation curve in any individual cell (Fig. 5a), indicating that PtdIns(4,5)P$_2$ is obligatory for effective Ca$^{2+}$ transport by PMCA2 in the kilohertz range.

When we analysed our PMCA2–NPTN structures, we found a density corresponding to PtdIns(4,5)P$_2$ in all but the E2.P$_i$ state (Fig. 1). The PtdIns(4,5)P$_2$-binding site is located in a pocket formed by the membrane-facing sides of helices M3, M5 and M7, in very close proximity to the Ca$^{2+}$-binding site (Fig. 5b–d and Extended Data Fig. 6a,b). The negatively charged head group of PtdIns(4,5)P$_2$ is stabilized by the positively charged vestibule of the pocket facing the cytoplasm, whereas the tail of the phospholipid interacts with hydrophobic residues in the interior of the binding pocket (Fig. 5b and Extended Data Fig. 6c,d). This tail is maintained at the same position between states, whereas the phosphate-rich head group appears more flexible (Fig. 5b–d, Extended Data Fig. 6a,b and Supplementary Videos 4 and 5). Ligand-coordination analyses echoed this by indicating that various residues in the PtdIns(4,5)P$_2$-binding pocket are able to coordinate the lipid at different stages of the transport cycle. Of these, interaction of the PtdIns(4,5)P$_2$ head group with residue Q351 appeared to be the most constant (Fig. 5b, Extended Data Fig. 6f and Supplementary Videos 4 and 5). Nonetheless, it is noticeable that the lipid tightly fits into the binding pocket across all states, indicating shape complementarity. In the E2.P$_i$ state, however, the conformation of the PtdIns(4,5)P$_2$-binding pocket becomes markedly wider and therefore disfavours high-affinity PtdIns(4,5)P$_2$ binding to an extent that a double-tailed lipid temporarily

replaces it in our purified sample (Extended Data Fig. 6d,e). In summary, there are four features of the PtdIns(4,5)P$_2$-binding pocket that render it specific for this particular lipid: shape complementarity, a positively charged vestibule, a hydrophobic patch in the interior and direct interaction of Q351 with a phosphate group of PtdIns(4,5)P$_2$.

In the E1 states, where PtdIns(4,5)P$_2$ binding is particularly tight, an oxygen of the fatty acid chain ester bond of PtdIns(4,5)P$_2$ interacts with residue N841 of PMCA2 (Fig. 5c,d, Extended Data Figs. 6a,b and 7a and Supplementary Videos 4 and 5). This interaction stabilizes N841 in the midst of a hydrophobic environment, thereby stabilizing this state like a latch that prevents interference of N841 with the neighbouring Ca$^{2+}$-binding site. The equivalent residue (N768) in SERCA1a is involved in the Ca$^{2+}$ binding at site I (Extended Data Fig. 7b,c), which would prevent this pump from interacting with PtdIns(4,5)P$_2$. At the same time, S710 in SPCA1, the residue corresponding to N841 in PMCA2, is also poorly suited to be a PtdIns(4,5)P$_2$ interactor due to the short length of its side chain (Extended Data Fig. 7d).

Although the PtdIns(4,5)P$_2$–N841 interaction is maintained throughout all E1 states, it is disrupted by the large movements that occur during the transition from the E1-Ca-ATP to the E2P state. There, the binding pocket is widened and N841 is reoriented and interacts with Q837 instead of PtdIns(4,5)P$_2$, explaining the profoundly weaker binding of PtdIns(4,5)P$_2$ in the E2 states (Fig. 5c,d). Of note, this reorientation of N841 (Fig. 5d, Extended Data Figs. 6b and 7a and Supplementary Videos 4 and 5) is obstructed by the side chain of M872 in the E1 states. During transition into the E2P state (Fig. 5d, Extended Data Figs. 6b and 7a and Supplementary Videos 4 and 5), M872 rotates away from the PtdIns(4,5)P$_2$-binding site, thus allowing N841 to change its position and to interact with Q837.

In addition to the newly formed interaction with N841, Q837 in the E2P state maintains its interaction with D873, a residue that is directly involved in ion coordination in the Ca$^{2+}$-bound E1 states, but is displaced from the Ca$^{2+}$-binding site (Fig. 5d). The displacement of the D873–Q837 diad results in a change in the calculated pK$_a$ of D873 (from approximately 6.0 to approximately 7.0), which decreases the affinity of the residue for Ca$^{2+}$ binding and increases its affinity for protons (Supplementary Table 4 and Methods). Together, these processes establish direct structural and functional links between the binding sites for Ca$^{2+}$ and PtdIns(4,5)P$_2$, as well as the water-lined proton pathway. PtdIns(4,5)P$_2$ stabilizes the Ca$^{2+}$-bound states similar to a latch, whereas after disruption of the PtdIns(4,5)P$_2$–N841 interaction, the Ca$^{2+}$-free and protonated states are favoured.

To further analyse the crucial functions of N841 and the PtdIns(4,5)P$_2$-binding pocket in the catalytic cycle of PMCA2, we performed our cellular transport assay with the single and triple replacement mutants N841D and K334T–K347T–Q351L. The former should decrease PMCA2 affinity to PtdIns(4,5)P$_2$ by weakening interaction of the pump with the phospholipid tail, whereas the latter should do so by decreasing its binding to the phospholipid head group. In both cases, Ca$^{2+}$ transport of the mutated PMCA2–NPTN complexes was markedly reduced, close in efficiency to those observed with the mutants interfering with Ca$^{2+}$ binding or proton counter-transport (Fig. 5e). Furthermore, a similar reduction in transport efficiency was obtained with single-replacement mutants at positions Q837 and D873 (Fig. 5e). These results confirm the key role of the interplay between N841 and PtdIns(4,5)P$_2$ for ultrafast Ca$^{2+}$ pumping, as well as the requirement of positively charged and hydrophobic regions in the PtdIns(4,5)P$_2$-binding pocket. Together, our structural and functional analyses reveal that PtdIns(4,5)P$_2$ specifically binds to PMCA2 and, via N841, promotes both high-affinity binding and fast release of the transport substrate Ca$^{2+}$, making it essential for the proper operation of the pump (Fig. 5f).

## Pharmacological targeting

When we compared structures of SERCA1a with our PMCA2 structures, we found that the binding pocket for PtdIns(4,5)P$_2$ in PMCA2 is

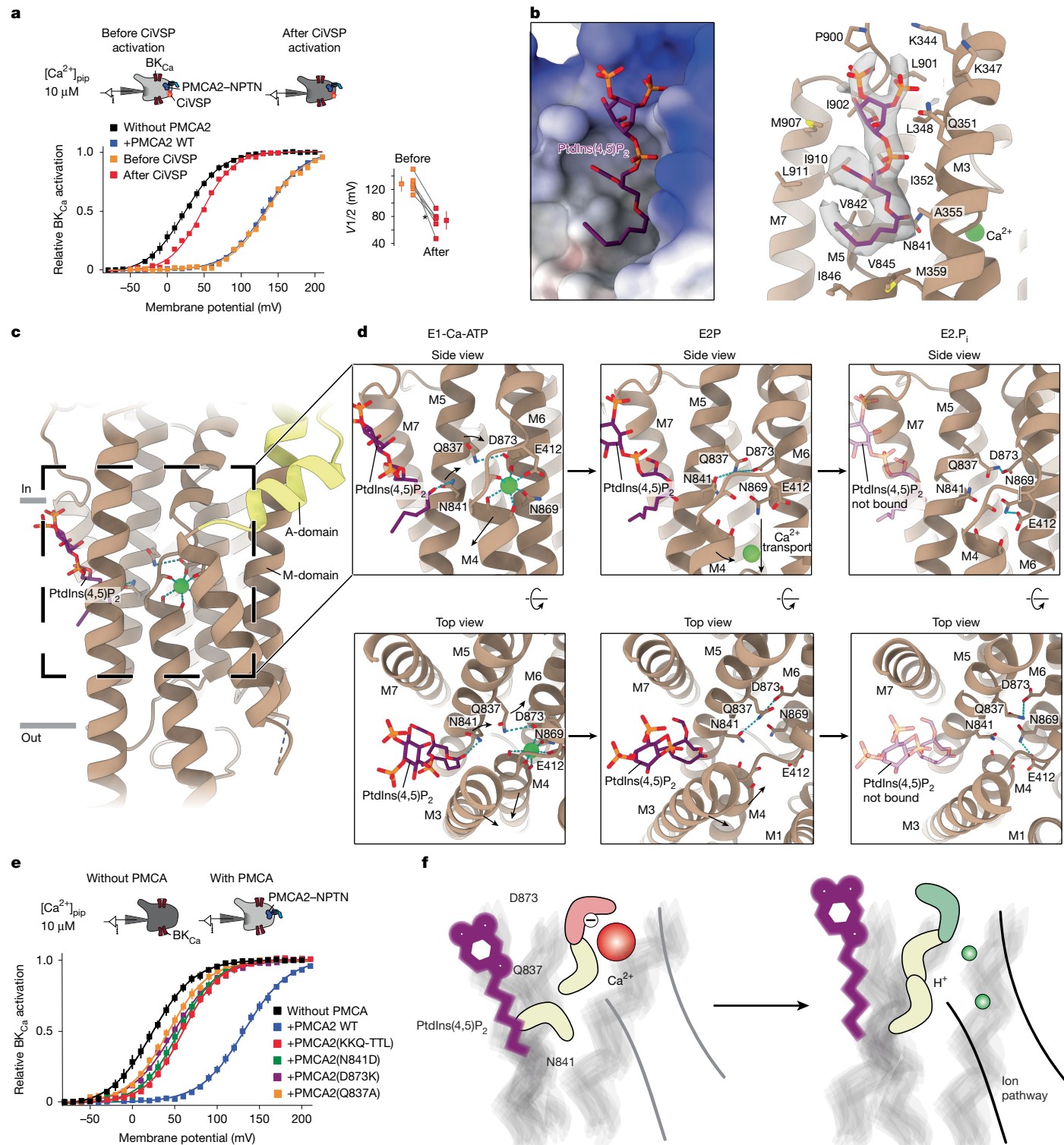

**Fig. 5 | Mechanism of PMCA activation by PtdIns(4,5)P₂. a**, $BK_{Ca}$ activation curves recorded as in Fig. 3 from CHO cells co-expressing PMCA–NPTN pumps before (orange) and after (red) activation of the PtdIns(4,5)P₂ phosphatase CiVSP (left). Activation curves in the absence and presence of PMCA2–NPTN (Fig. 3f) were added for comparison. A summary of the shifts observed in six individual cells is also shown. Data are mean ± s.e.m. The asterisk denotes the cell shown on the left. **b**, The PtdIns(4,5)P₂-binding site of PMCA2–NPTN in the E1-Ca state. A surface representation of PMCA2 coloured by electrostatic Coulomb potential ranging from −10 kcal (mol e)⁻¹ (red) to +10 kcal (mol e)⁻¹ (blue; left), and residues that form the lipid-binding pocket and the cryo-EM density of PtdIns(4,5)P₂ (right) are shown. **c**, Transmembrane region of PMCA2 in the E1-Ca-ATP state. The PtdIns(4,5)P₂-binding and Ca²⁺-binding sites are shown. **d**, Zooms of the connection between PtdIns(4,5)P₂-binding and Ca²⁺-binding sites in PMCA2–NPTN in the E1-Ca-ATP (left), E2P (middle) and E2.Pᵢ (right) states. Residues involved in Ca²⁺ binding are shown, as well as N841 and Q837 that mediate the interaction of PtdIns(4,5)P₂ with the Ca²⁺-binding site. The arrows indicate the direction of motion. PtdIns(4,5)P₂ is not bound in the E2.Pᵢ state, but it is shown semi-transparently to indicate its binding site. The top row depicts the protein models as side view, and the bottom row shows the models from the cytoplasmic side as top view. In the side views, membrane helices M1 and M3 are hidden for visualization purposes. See also Extended Data Figs. 6a,b and 7a and Supplementary Videos 4 and 5. **e**, $BK_{Ca}$ activation curves as in panel **a** but for PMCA–NPTN with the indicated mutants. Data are mean ± s.e.m. of 6–9 cells. **f**, Model highlighting the role of PtdIns(4,5)P₂ in the release of Ca²⁺ and counter-proton shuttling.

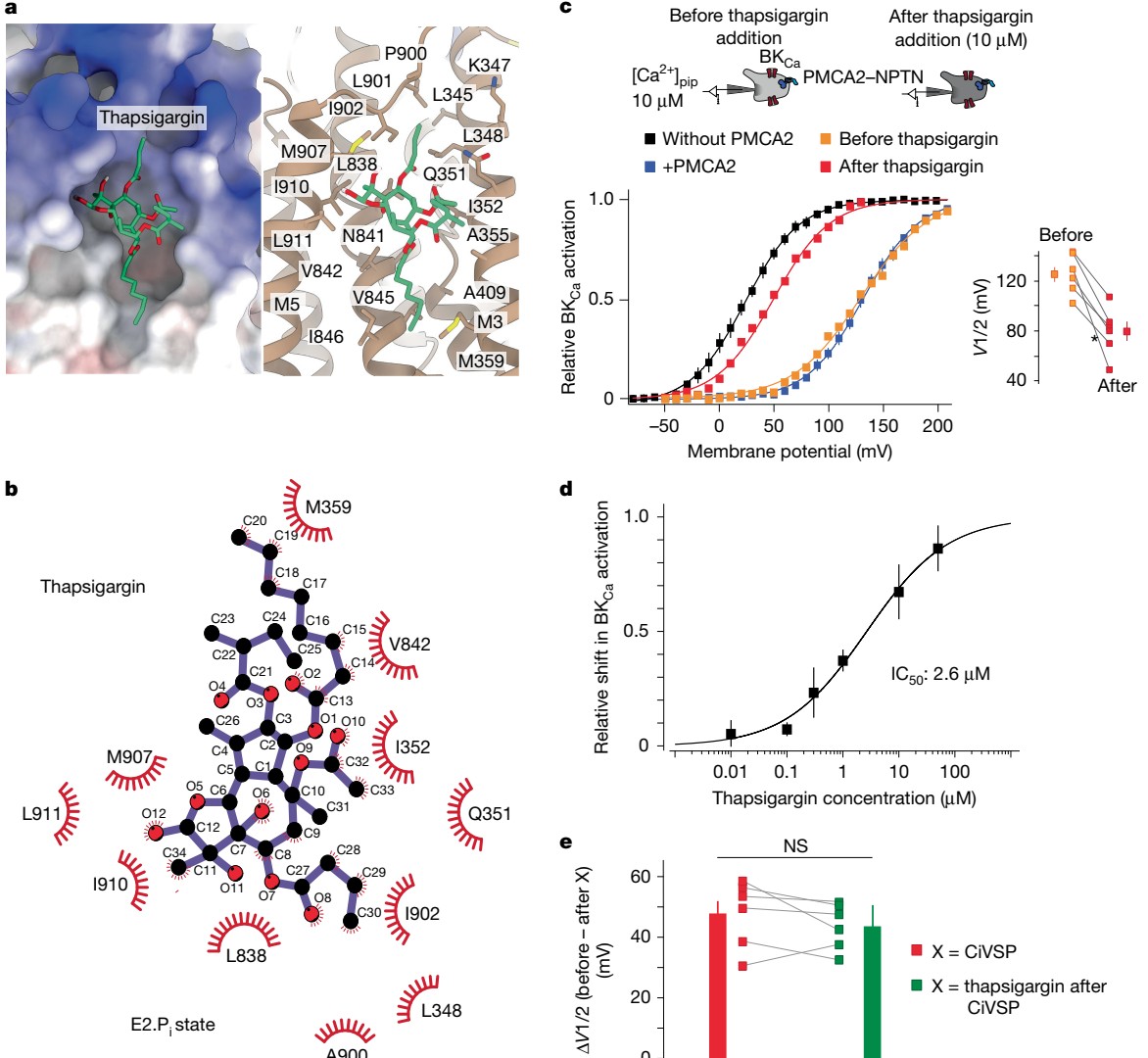

**Fig. 6 | Thapsigargin binds to the same modulatory pocket as PtdIns(4,5)P₂.**
**a**, The PtdIns(4,5)P₂-binding site of PMCA2 in the E2P state with docked thapsigargin, as a surface representation (left) coloured by electrostatic Coulomb potential ranging from −10 kcal (mol e)$^{-1}$ (red) to +10 kcal (mol e)$^{-1}$ (blue), and as a molecular model (right) highlighting potentially interacting PMCA2 residues. **b**, Molecular docking of thapsigargin to the E2.P$_i$-state structure of PMCA2 confidently places the inhibitor into the modulatory pocket of PMCA2. **c**, Ca$^{2+}$ transport activity of PMCA2–NPTN pumps is strongly inhibited by thapsigargin. BK$_{Ca}$ activation curves recorded as in Fig. 3 from a CHO cell co-expressing PMCA2–NPTN before (orange) and after (red) application of 10 μM thapsigargin (left). Activation curves in the absence and presence of PMCA2–NPTN (from Fig. 3f) were added for comparison.

A summary of shifts observed in six individual cells is also shown. Data are mean ± s.e.m. **d**, Dose–response curve for PMCA2 inhibition by thapsigargin determined from the relative shift of BK$_{Ca}$ activation in experiments as in panel **c**; data are mean ± s.e.m. of 5–10 experiments. The line is the result of a Hill fit with values for IC$_{50}$ and Hill coefficient of 2.6 μM and 0.6. **e**, After PtdIns(4,5)P₂ depletion by CiVSP, thapsigargin fails to inhibit pump activity of PMCA2–NPTN complexes. The shift in BK$_{Ca}$ activation determined in six CHO cells after activation of CiVSP alone (squares in red) and after 10 μM thapsigargin application following CiVSP activation (squares in green). The bars are mean ± s.e.m. Data are not significantly different (NS; two-sided Mann–Whitney *U*-test).

equivalent to that of thapsigargin in SERCA1a (Fig. 6a and Extended Data Fig. 7e,f). Thapsigargin is a potent plant-derived inhibitor of SERCA and, upon binding, locks the pump in the Ca$^{2+}$-free E2 state[48]. Although the more hydrophilic nature of PtdIns(4,5)P₂ shifts its binding site slightly towards the membrane–cytoplasm interface, the equivalents of many PtdIns(4,5)P₂-binding residues in PMCA2 participate in thapsigargin binding in SERCA1a (Extended Data Fig. 7e,f). In addition, molecular docking of thapsigargin to PMCA2 in the E2.P$_i$ state demonstrated that the inhibitor readily occupies the PtdIns(4,5)P₂-binding site (Fig. 6a,b).

To test whether thapsigargin actually affects PMCA2, we used thapsigargin in our BK$_{Ca}$-based transport assay and determined activation curves before and after bath application of the drug at various concentrations. The results showed that thapsigargin indeed inhibited

the pumping activity of PMCA2–NPTN in a concentration-dependent manner with a half-maximal inhibition (IC$_{50}$) of 2.6 μM (Fig. 6c,d). Moreover, BK$_{Ca}$ activation curves recorded in individual cells before and after application of 10 μM thapsigargin indicated inhibition of Ca$^{2+}$ transport with efficiency similar to CiVSP-mediated PtdIns(4,5)P₂ depletion, albeit with some cell-to-cell variation (Figs. 5a and 6c). This prompted us to explore whether thapsigargin-mediated PMCA2 inhibition may be PtdIns(4,5)P₂ dependent. When we applied the drug after PtdIns(4,5)P₂ depletion via CiVSP, thapsigargin failed to exert any additional inhibitory effect on PMCA2 (Fig. 6e), indicating that the effects of thapsigargin and PtdIns(4,5)P₂ depletion are not cumulative and that thapsigargin-mediated inhibition probably results from competition with PtdIns(4,5)P₂ for the same binding pocket. Together,

these results suggest that the PtdIns(4,5)P$_2$-binding site represents a druggable site for interfering with the transport activity of PMCA-type Ca$^{2+}$-ATPases and thus for modulating intracellular Ca$^{2+}$ levels.

## Discussion

The eight different structures of PMCA2 and PMCA2–NPTN complexes together with the functional recordings presented in this study provide detailed insight into the transport cycle of this classical plasma membrane Ca$^{2+}$-pump (Supplementary Video 1). They revealed the binding sites for the accessory subunit NPTN and the phospholipid PtdIns(4,5) P$_2$, and elucidated the structural features and mechanisms promoting the uniquely fast kinetics and the fundamental role of the phospholipid for the Ca$^{2+}$-proton counter-transport of these primary transporters.

The first unique feature of PMCA is the high-affinity binding of Ca$^{2+}$ to the sole substrate-binding site. This process is independent of the binding of the nucleotide ATP, which accelerates entry into the transport cycle (Fig. 1). The second remarkable property is the small conformational changes observed for the cytoplasmic domains of PMCA2 associated with the transitions between the states of the transport cycle compared with other Ca$^{2+}$-ATPases. This includes the rate-limiting step from E1-Ca-ATP to E2P that is driven by ATP hydrolysis, which appeared to be too dynamic to obtain a stable intermediate state with Ca$^{2+}$ and ADP bound (E1P-Ca-ADP). The third striking feature is the noticeably small interaction interfaces between the cytoplasmic domains in all states of the transport cycle (Supplementary Table 3). We propose that the resulting decrease in energy required for conformational changes immediately promotes an increased speed of the state-to-state transitions. The fourth and most important feature is the long-sought interaction of PMCA2 with the phospholipid PtdIns(4,5)P$_2$. The lipid binds to a pocket close to the Ca$^{2+}$-binding site. The pocket is ideally positioned to promote both high-affinity binding of Ca$^{2+}$ to its sole binding site in the E1 states and to facilitate its subsequent release, enabling the pump to shift to the E2P state (Figs. 1b and 5c,d). These actions are mediated by a 'latch-like' action of the lipid (Fig. 5f): in the E1 states, PtdIns(4,5)P$_2$-mediated stabilization of N841 prevents interaction with the 'Q837–D873 bracket' that is crucial for coordinated Ca$^{2+}$ binding, with D873 in a deprotonated negatively charged state (due to a pK$_a$ of about 6.0). Following ATP-driven conformational rearrangements, the interaction between PtdIns(4,5)P$_2$ and N841 is disrupted and N841 binds to the Q837–D873 diad instead. As a consequence, Ca$^{2+}$ is released and counter-protons can enter. This process is supported by a change in the pK$_a$ of D873 from the slightly acidic to the neutral range (pK$_a$ of about 7.0), similar to that described in SERCA[49].

The fundamental importance of PtdIns(4,5)P$_2$ for PMCA–NPTN function is evidenced by its depletion from the plasma membrane under intact cellular conditions via CiVSP, a PtdIns(4,5)P$_2$-specific phosphatase that completely abolished transport activity (Fig. 5). Mechanistically, the effect of CiVSP results from removal of the negatively charged 5' head group of the phospholipid, which, by binding to the positively charged vestibule of the PtdIns(4,5)P$_2$ pocket, forms one of the four structural determinants defining interaction of PtdIns(4,5) P$_2$ with PMCA2. The other determinants are shape complementarity, a hydrophobic patch and the fatty acid ester bond of PtdIns(4,5)P$_2$ that interacts with the latch-like operating N841 in PMCA2 (Fig. 5). Similarly, at least two-tiered interactions have been identified for PtdIns(4,5)P$_2$ in structures of various membrane proteins, mainly ion channels and transporters. In most cases, the negatively charged phosphate groups of the lipid interact with positively charged arginine/ lysine residues inducing conformational rearrangements (mostly in voltage-sensing domains and associated linkers) that either favour or disfavour active/open states of the respective protein[50–56]. The fatty acid tails of PtdIns(4,5)P$_2$ are more implied in anchoring or correct placing of the phospholipid rather than in controlling functional activity as identified here for the PMCA-mediated Ca$^{2+}$ transport[50,51,55].

In addition to its crucial importance for normal PMCA operation, the PtdIns(4,5)P$_2$-based activation mechanism also represents an exquisite avenue for pharmacological interference. This was showcased by inhibiting PMCA activity with thapsigargin, a compound that competes with PtdIns(4,5)P$_2$ at its binding site (Fig. 6). These results provide a proof of principle for the development of small-molecule modulators that increase [Ca$^{2+}$]$_i$, prolong Ca$^{2+}$ signals or induce cell death similarly to thapsigargin-derived prodrugs[57–60].

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

## Methods

### Cloning and cell culture

Mouse NPTN (EMM1002-204751096, Dharmacon; GenBank accession NM_009145.2) and PMCA2 (MG223661, OriGene; accession NM_001036684.3) were subcloned in pcDNA3.1[+]. Constructs of PMCA2 and PMCA2(N841D) were generated with a C-terminal FLAG-tag (sequence added: ENLYFQGGDYKDDDDK), another construct of PMCA2 contained a C-terminal His-FLAG epitope tag (sequence added: ENLY FQGGHHHHHHHHHGDYKDDDDK), and the NPTN expression construct was tagged with a C-terminal His-tag (sequence added: ENLYFQGGH HHHHHH). All constructs were verified by sequencing and prepared as separate cDNAs for transient (co)transfection of tsA201 *NPTN/BSG* double knockout cells[19], CHO wild-type or CHO *Atp2b2-4*[−/−] cells.

For quantification purposes in the electrophysiological experiments, PMCA2, PMCA2(E412K), PMCA2(Q837A), PMCA2(D873K), PMCA2(S877F), PMCA2(KKQ344,347,351TTL) and PMCA2(N841D) were C-terminally fused with GFP (in pCMV6), whereas NPTN and CiVSP[61] were C-terminally fused with RFP (in pRFP-N1). The $BK_{Ca}$ used had the sequence of UniProt ID Q08460. All constructs were verified by sequencing and prepared as separate cDNAs. CHO and tsa201 cells cultivated at 37 °C and 5% $CO_2$ were transiently (co)transfected with the indicated cDNAs via the JetPEI reagent and measured 2–4 days after transfection.

The first Ig-like domain of NPTN encompassing residues 29–148 (Ig1) was cloned into a pEG BacMam vector with an Igκ sequence as signal peptide.

### Electrophysiological measurements of PMCA2 activity

Whole-cell patch-clamp recordings were performed at room temperature using a HEKA EPC 10 amplifier. Currents were low-pass filtered at 3–10 kHz and sampled at 20 kHz. Leak currents were subtracted with a P/4 protocol at a holding potential of −90 mV. Serial resistance was 50–70% compensated using the internal compensation circuitry. The standard extracellular solution contained 5.8 mM KCl, 144 mM NaCl, 0.9 mM $MgCl_2$, 1.3 mM $CaCl_2$, 0.7 mM $NaH_2PO_4$, 5.6 mM D-Glc and 10 mM HEPES pH 7.4. Recording pipettes pulled from quartz glass had resistances of 2.5–4.5 MΩ when filled with internal solution.

For recordings of $BK_{Ca}$ currents, intracellular solution contained 139.5 mM KCl, 3.5 mM $MgCl_2$, 2 mM DiBrBAPTA, 5 mM HEPES, 2.5 mM $Na_2ATP$ and 0.1 mM $Na_3GTP$ (pH 7.3); $CaCl_2$ was added to obtain the following free $[Ca^{2+}]_{pip}$: 100 nM, 1 μM, 10 μM and 50 μM. WEBMAXC STANDARD (https://somapp.ucdmc.ucdavis.edu/pharmacology//bers/max-chelator/webmaxc/webmaxcS.htm) was used for calculation of the appropriate amount of calcium to be added to the internal solution, and final free $Ca^{2+}$ concentrations were checked with a $Ca^{2+}$-sensitive electrode (World Precision Instruments). Steady-state activation of $BK_{Ca}$ channels at distinct $[Ca^{2+}]_i$ was determined using test pulses ranging from −80 mV to +200 mV (in 10-mV increments), followed by a repolarization step to 0 mV. Conductance–voltage relations were determined from tail current amplitudes measured 0.5 ms after repolarization to the fixed membrane potential and normalized to maximum. Data were fitted with a Boltzmann function: $g/g_{max} = g_{max}/(1 + \exp((V_h - V_m)/k))$, where $V_h$ is the voltage required for half-maximal activation and $k$ is the slope factor. The $PtdIns(4,5)P_2$-specific 5′-phosphatase CiVSP was activated by membrane depolarization to 60 mV before the $BK_{Ca}$ current recordings. All chemicals except for DiBrBAPTA (Alfa Aesar) and thapsigargin (Tocris) were purchased from Sigma. Stock solutions of 5 mM and 50 mM thapsigargin were prepared in DMSO.

### Quantification of PMCA2 on cell membranes by immunogold labelling

For determination of the PMCA2 density in tsA cells, the SDS-FRL technique was used as previously described[62] with some modifications. Cells were transiently transfected with the aforementioned plasmids (coding for PMCA2–GFP and NPTN–RFP) or an empty vector with Jet-PEI transfection reagent according to the manufacturer's instructions. After 2 days of incubation, cells were rinsed with 25 mM PBS and then fixed with 4% paraformaldehyde (Roth) in PBS for 10 min. Cells were harvested by scraping, pelleted by centrifugation (500*g* for 5 min); cell pellets were sandwiched between copper carriers for high-pressure freezing (HPM100, Leica). Frozen pellets were then fractured into two parts at −120 °C and the fractured facets were coated with carbon (5 nm), platinum-carbon (2 nm) and an additional layer of carbon (20 nm) in a freeze-fracture replica machine (ACE900, Leica). Replicas were digested at 60 °C in a solution containing 2.5% SDS and 20% sucrose diluted in 15 mM Tris buffer (pH 8.3) for 48 h followed by 37 °C for 20 h. The replicas were washed in washing buffer comprising 0.05% bovine serum albumin (BSA; Roth) and 0.1% Tween-20 (Roth) in 50 mM Tris-buffered saline (TBS) and then blocked in a solution containing 5% BSA and 0.1% Tween-20 in TBS at room temperature for 1 h. Subsequently, replicas were incubated at 15 °C for two overnight periods in a TBS solution containing 1% BSA and 0.1% Tween-20 and PMCApan (5F10, mouse /Ms, 1:3,000; MA3-914, Invitrogen). Replicas were washed in washing buffer and then reacted with 6 nm (Rb) or 12 nm (Ms or Rb) gold particle-conjugated secondary antibodies (1:60; Jackson ImmunoResearch Laboratories) at 15 °C overnight. Finally, replicas were washed in TBS followed by distilled water, mounted on Formvar-coated 100 mesh grids, and analysed with transmission electron microscopes (CM100, Philips or Zeiss LEO 906 E). As the PMCA-pan antibody targeted an intracellular epitope, immunoreactivity was observed on the protoplasmic face (P-face) of the plasma membrane. The density of immunoparticles labelling a protein (or proteins) of interest was calculated by dividing the absolute number of gold particles by the surface area of CHO cells, dendritic shafts of Purkinje cells and varicosities of parallel fibres[32,63].

### Complexome profiling

For analysis of PMCA-associated, SERCA-associated and SPCA-associated complexes (Extended Data Fig. 4c,d), endoplasmic reticulum-enriched membrane fractions[64] were prepared from freshly isolated adult mouse brains and gently solubilized with CL-47 (salt replaced by 750 mM aminocaproic acid). After ultracentrifugation, 1 mg of solubilized protein was concentrated by ultracentrifugation into a 20%/50% sucrose cushion supplied with 0.125% Coomassie G250 Blue. The sample was then subjected to high-resolution cryo-slicing blue native gel mass spectrometry (csBN-MS) as recently described[65]. In brief, the blue native gel (1–18%) lane was sliced into 359 samples (0.3-mm intervals), each digested with trypsin, and the obtained peptides (dissolved in 0.5% (v/v) trifluoroacetic acid) were measured on a QExactive mass spectrometer coupled to an UltiMate 3000 RSLCnano HPLC system (Thermo Scientific). After absorption on a C18 PepMap100 precolumn (300 μm i.d. x 5 mm; particle size of 5 μm; 0.05% (v/v) trifluoroacetic acid; 5 min at 20 μl min[−1]), peptides were eluted with an aqueous-organic gradient (eluent A: 0.5% (v/v) acetic acid; eluent B: 0.5% (v/v) acetic acid in 80% (v/v) acetonitrile) as follows: 5 min 3% B, 120 min from 3% B to 30% B, 20 min from 30% B to 50% B, 10 min from 50% B to 99% B, 5 min 99% B, 5 min from 99% B to 3% B, and 10 min 3% B (flow rate of 300 nl min[−1]). Separation column was a SilicaTip emitter (i.d. 75 μm; tip of 8 μm; New Objective) packed manually (23 cm) with ReproSil-Pur 120 ODS-3 (C18; particle size of 3 μm; Dr. Maisch HPLC) from which eluting peptides were directly electrosprayed (2.3 kV; transfer capillary temperature of 300 °C) in positive ion mode. Mass spectrometry acquisition parameters were: maximum tandem mass spectrometry injection time of 400 ms; dynamic exclusion duration of 60 s; minimum signal/intensity threshold of 40,000 (counts), top 15 precursors fragmented; and isolation width of 1.4 $m/z$.

Peak lists were obtained from fragment ion spectra using msconvert (part of ProteoWizard; http://proteowizard.sourceforge.net/; v3.0.11098). Mass offsets were corrected by linear shifting (±50 ppm

pre-search), and precursors finally searched with 5 ppm mass tolerance against all mouse, rat and human entries of the UniProtKB/Swiss-Prot database (release 20181205). Acetyl (protein N terminus), carbamido-methyl (C), Gln→pyro-Glu (N-terminal Q), Glu→pyro-Glu (N-terminal E), oxidation (M), phospho (S, T, Y) and propionamide (C) were chosen as variable modifications (fragment mass tolerance of 20 mmu). One missed tryptic cleavage was allowed. The expected value cut-off for peptide assignment was set to 0.5. Related identified proteins (subset or species homologues) were grouped using the name of the predominant member. Proteins either representing exogenous contaminations (for example, keratins, trypsin and IgG chains) or identified by only one specific peptide were not considered. Computation of protein abundance profiles from peptide signal intensities was done using MaxQuant and in-house developed software as previously described[65]. Slice numbers were converted to apparent complex molecular weights by the sigmoidal fitting of log(MW) of marker complexes versus their observed profile peak maxima slice index. Protein abundance profiles were finally smoothed using a Gaussian filter (width set to 1.4).

## Protein purification

tsA201 *NPTN/BSG* double knockout cells (co)transfected with PMCA2 and NPTN-encoding plasmids were grown in 1 l suspension culture and harvested 72 h post-transfection by centrifugation. For two-step affinity purification (all steps carried out at 4 °C), 6–12 g batches of cells were first solubilized with ComplexioLyte 145 detergent buffer (Logopharm) supplemented with EDTA + EGTA (Supplementary Table 1) and protease inhibitors (1.5 mM iodoacetamide, leupeptin, pepstatin, aprotinin (all 10×, according to the manufacturer's recommendation) and 1 mM PMSF) at 8–9 ml g$^{-1}$ of cell pellet using a glass potter (tight pestle) and 1-min incubation in an ultrasonic bath. After 20 min, MgCl$_2$ was added and the homogenate ultracentrifuged at 150,000*g* for 30 min. Then, imidazole was added and the solubilizate mixed with 1.0–1.6 ml of solubilization buffer-equilibrated Ni-Sepharose beads (Cytiva). After incubation for 1.5 h on a tumbling shaker, the beads were sedimented by brief centrifugation and transferred to an Econo-Column 1.0 × 5 cm (Bio-Rad) operated by a peristaltic pump. Washing was performed for 40 min at a flow rate of 0.6 ml min$^{-1}$ using washing buffer WP1 (20 mM Tris-HCl pH 7.4, 150 mM NaCl, imidazole and glyco-diosgenin (GDN); Supplementary Table 1) containing protease inhibitors (1.5 mM iodoacetamide, leupeptin, pepstatin, aprotinin (all 1×) and 0.5 mM PMSF). Bound proteins were eluted with 5–6 ml WP1/230 mM imidazole for 30 min and added to 0.8–1.4 ml WP1-equilibrated anti-FLAG M2 affinity gel (A2220, Sigma). After incubation for 2 h on a tumbling shaker, beads were again sedimented by brief centrifugation, transferred to an Econo-Column 1.0 × 5 cm (Bio-Rad) operated by a peristaltic pump, and thoroughly washed for 40–60 min with 30 ml WP2 (20 mM Tris-HCl pH 7.4, 150 mM NaCl and GDN; Supplementary Table 1). For elution, 350–600 µg 3× FLAG peptide (Sigma) were dissolved in 1.2–2 ml WP2 and incubated or eluted in batches over 45 min. After determination of the protein concentration by Nanodrop (Thermo Fisher Scientific) and Bradford assay (Bio-Rad), the eluate was concentrated (VivaSpin 6 with 100,000 kDa MWCO or Centricon with 100,000 kDa MWCO) to 2–4 mg ml$^{-1}$, then ultracentrifuged for 15 min at 100,000*g*. The final protein concentration was adjusted to 1.2–2.5 mg ml$^{-1}$. Aliquots of the eluates and final concentrates were taken for quality control by SDS–PAGE and BN–PAGE. The specific modifications, including addition of co-factors and stabilizers, that were made for individual preparations are listed in Supplementary Table 1.

For expression of the Ig1 of NPTN, HEK293 GnTI$^-$ cells (negative mycoplasma test; CRL-3022, American Type Culture Collection) were transiently transfected with the expression plasmids using polyethylenimines (Polysciences). Approximately 1 mg plasmids were pre-mixed with 3 mg polyethylenimines in 50 ml fresh medium for 30 min before transfection. For transfection, 50 ml of this mixture was added to 2-l cell cultures and incubated for 30 min. The transfected cells were grown in

suspension in FreeStyle medium (Gibco Life Technologies) at 37 °C for 48 h before harvesting. Ig1, secreted into the medium, was harvested after pelleting the cells by centrifugation at 1,500*g* for 10 min. The supernatant was applied using a peristaltic pump set to a flow rate of 0.5 ml min$^{-1}$ to two 5-ml HisTrap Excel columns (Cytiva) connected in tandem. The columns were washed successively with 5 mM and 20 mM imidazole containing buffer (20 mM Tris (pH 8.0) and 150 mM NaCl). Finally, the protein was gradient eluted on a Bio-Rad NGC machine using the same buffer containing 0–500 mM imidazole. The eluted protein was further purified by gel-filtration chromatography using a Superose 12 10/300 column (Cytiva), concentrated to 3.6 mg ml$^{-1}$ and used for crystallization. For the latter, the JSG Core I-IV, Pact, PEG suite and Protein Complex suites were utilized (Qiagen).

## Crystallization of the NPTN N-terminal Ig1 domain

To crystallize the first Ig-like domain (Ig1) of NPTN, the sitting-drop vapour diffusion method was used. Crystals grew in 0.1 M HEPES (pH 7.5) and 30% (v/v) PEG 400 and were directly flash frozen in liquid nitrogen after harvesting. Diffraction data were collected at 100 K on beamline X10SA at the Swiss Light Source (Paul Scherrer Institute).

## NPTN N-terminal Ig1 domain crystal data processing and refinement

Data were collected on a single crystal that diffracted to 2 Å. Diffraction data were integrated and scaled with XDS[66]. The space group was determined to be P3$_2$21 with cell parameters of $a = b = 85.907$ Å and $c = 163.285$ Å. The AlphaFold model of mouse NPTN (ID: AF-P97300-F1) was used for molecular replacement using the Phaser program[67] in PHENIX[68]. Four copies of the Ig1 domain per unit cell gave the best solution. Refinement and model building were carried out in PHENIX and Coot[69], respectively. The final data statistics have been provided in Extended Data Table 2. The refined model has Ramachandran-favoured values of 97.40%, with 2.16% and 0.413% in the allowed and outlier region, respectively.

## Cryo-EM data acquisition

After using negative-stain analysis to identify optimal conditions for downstream processing of the PMCA–NPTN complex, all samples were directly screened at cryogenic temperatures using an Talos Arctica microscope (Thermo Fisher Scientific) operated at 200 kV. For this, 3–4 µl of protein solution (1.5–2.5 mg ml$^{-1}$) was applied onto freshly glow-discharged R1.2/1.3 300 holey carbon grids (Quantifoil), blotted at 4 °C and 100% humidity by a Vitrobot Mark IV (Thermo Fisher Scientific) using a 3.0-s blot time and blot force of 0, and subsequently vitrified in liquid nitrogen-cooled liquid ethane.

Datasets were collected using EPU software on an X-FEG equipped Titan Krios G3 microscope (Thermo Fisher Scientific) operated at 300 kV. All datasets were collected using the aberration-free image shift feature of EPU (Thermo Fisher Scientific) that was used to speed up the data collection process. Equally dosed frames were collected using a K3 direct electron detector (Gatan) in super-resolution mode in combination with a BioQuantum energy filter (Gatan) set to a filter width of 20 eV or 15 eV. The details of all eight datasets including pixel size, total electron exposure, exposure time, number of frames and defocus range are summarized in Extended Data Table 1. Data collection was monitored live using TranSPHIRE[70], allowing for on-the-fly adjustments of data acquisition settings such as defocus range or astigmatism when necessary. The total number of images collected is summarized in Extended Data Table 1.

## Cryo-EM data processing

Within TranSPHIRE, data preprocessing included drift and gain correction with MotionCor2 (ref. 71) to create aligned, dose-weighted micrographs. The super-resolution images were binned twice after motion correction to speed up further processing steps. Contrast transfer

function estimation was also performed within TranSPHIRE using CTFFIND (v4.1.10)[72]. After data collection, unaligned frame averages were manually inspected, and bad quality images (due to, for example, ice contamination) were removed from the datasets.

All datasets were processed using the same general strategy. Single particles were picked automatically with crYOLO using the general model[73]. The particles were then extracted in a box of 300 × 300 pixels. Reference-free 2D classification and cleaning of the dataset were performed with the iterative stable alignment and clustering approach ISAC[74] in SPHIRE[75]. ISAC processing was performed at a pixel size of 3.59 Å per pixel. A subset of particles producing 2D class averages and reconstructions with high-resolution features were then selected for further processing in CryoSPARC[76] or RELION (3.1)[77]. In the case of PMCA–NPTN in the E1-Ca state, the initial model and refinement were carried out in SPHIRE using RVIPER and MERIDIEN, respectively. For other states, the initial model was obtained with the ab initio module of cryoSPARC. The particles were then further sorted using 3D classifications in RELION or heterogenous refinements in cryoSPARC. For the reconstruction of PMCA–NPTN in the E1-Ca state, additional focused 3D classifications in RELION were carried out using masks that focused on the cytoplasmic domains of PMCA or the extracellular regions of NPTN. The latter classification yielded a particle set that revealed density for the Ig1 domain of NPTN. For all states, further Bayesian polishing and contrast transfer function refinement were then also carried out in RELION. The final round of high-resolution non-uniform refinement for all reconstructions was done in cryoSPARC.

## Model building and refinement of cryo-EM structures

The density map of PMCA2–NPTN in the E2.P$_i$–AlF$_4$ state was of high quality and enabled accurate modelling. We therefore modelled this state first. To build the structure, the previously reported model of the human PMCA1–NPTN complex (Protein Data Bank (PDB) ID: 6A69)[16] was fitted into the density map of PMCA in the E2.P$_i$ state as a rigid body using UCSF Chimera[78]. The model was then mutated to match mouse PMCA2 and mouse NPTN. PDB 6A69 only comprises the structure of the transmembrane helix and third Ig-like domain (Ig3) of NPTN. We therefore used an AlphaFold2 prediction (ID: AF-P97300-F1)[79] as the initial template for modelling the second Ig-like (Ig2) domain of NPTN. After fitting all components, the constructed model was further adjusted to fit the density using Coot[69]. For the glycosylation sites on Ig2 and Ig3 of NPTN, only the asparagine-linked NAG moiety was modelled, because the remaining regions of the sugars were of lower quality. Modelling of the glycosylation patterns was guided by the unsharpened cryo-EM maps. The structure was then iteratively refined using a combination of real-space refinement in PHENIX[68] and model adjustment in Coot until convergence as evaluated by model-to-map fit with valid geometrical parameters. This structure was then used as the initial model for the modelling of all other states.

To model the other states, the structure of PMCA2–NPTN in the E2.P$_i$–AlF$_4$ state was fitted in the corresponding density maps. Because the cytoplasmic domains of PMCA2 move substantially throughout the Post–Albers cycle, the A-domain and N-domain had to be fitted separately as rigid bodies in some of the structures. We used Loc-Spiral map sharpening[80] to guide the fitting and modelling of the more poorly resolved A-domain and N-domain regions. The models were then adjusted by manual model building in Coot. The extra density corresponding to a lipid moiety was modelled as a PtdIns(4,5)P$_2$ molecule (PDB ligand ID: KXP) in all the structures except the AlF$_4$-bound PMCA2–NPTN complex (E2.P$_i$ state), where the corresponding density was not clearly attributable to PtdIns(4,5)P$_2$. The models were then further refined using a combination of model adjustment in Coot and real-space refinement in PHENIX[68]. Validation statistics computed by PHENIX using MolProbity[81] were used to validate the overall geometry of the models.

For the N-terminal Ig-like domain (Ig1) of NPTN, the density was poorly resolved in most of the reconstructions, indicating flexibility.

However, as mentioned above, the cryo-EM density map of PMCA–NPTN in the E1-Ca state revealed density for the Ig1 domain following a focused 3D classification approach in RELION (Supplementary Fig. 1). This map was uploaded to the Electron Microscopy Data Bank as a separate entry (EMD-51625). Because the local resolution of Ig1 was still too low to allow for de novo modelling, we solved its structure using X-ray crystallography (see above) and rigid-body fitted the structure into the density map. Through this approach, a complete atomic model of NPTN could be constructed. Hence, the structure of PMCA–NPTN in the E1-Ca state that was uploaded to the PDB contains all three Ig-like domains of NPTN (PDB ID: 9GTB). For all other states of the PMCA–NPTN complex, only the Ig2 and Ig3 domains of NPTN are included in the uploaded models.

All figures displaying cryo-EM densities and protein structures were prepared in UCSF ChimeraX[82].

## Data analysis

Root-mean-square deviations between different states of the various Ca$^{2+}$-ATPases were calculated using the Matchmaker function of UCSF ChimeraX. The output values across all atom pairs were reported. Buried area analysis was performed using the PISA server[83,84]. Docking of thapsigargin to PMCA2 was done using HADDOCK[85,86], with all surface residues of the PMCA2 transmembrane domain being indicated as possible active site residues. From the four best structures of the top-scoring cluster (representing 73% of all water-refined models generated), one most closely resembling the empirically established conformation of thapsigargin bound to SERCA1a was selected for figure-making purposes. The extent of PtdIns(4,5)P$_2$ and thapsigargin interactions with PMCA2 and SERCA1a were calculated using LigPlot$^{+}$[87]. pK$_a$ values were calculated using PROPKA[88]. Docking of thapsigargin to PMCA2 was done using HADDOCK[85]. The residue displacement plot shown in Fig. 2c was generated using a custom script obtained from GitHub (https://github.com/schaefer-jh/motionviz). The arrows indicate Cα atom movement between the E1-Ca-ATP and EP2 states of PMCA2.

## Reporting summary

Further information on research design is available in the Nature Portfolio Reporting Summary linked to this article.

## Data availability

The cryo-EM maps were deposited in the Electron Microscopy Data Bank with accession ID (dataset in brackets): EMD-51545 (PMCA–NPTN E1 state), EMD-51546 (PMCA–NPTN E1-ATP state), EMD-51560 (PMCA–NPTN E1-Ca state), EMD-51625 (PMCA–NPTN E1-Ca state used to fit in NPTN Ig1), EMD-51548 (PMCA–NPTN E1-Ca-ATP state), EMD-51544 (PMCA–NPTN E2P state), EMD-51547 (PMCA–NPTN E2.P$_i$ state), EMD-51558 (PMCA-alone E2P state) and EMD-51549 (PMCA-alone E1-Ca-ATP state). The atomic coordinates were deposited in the PDB under the accession ID (dataset in brackets): 9GSE (PMCA–NPTN E1 state), 9GSF (PMCA–NPTN E1-ATP state), 9GTB (PMCA–NPTN E1-Ca state), 9GSH (PMCA–NPTN E1-Ca-ATP state), 9GSD (PMCA–NPTN E2P state), 9GSG (PMCA–NPTN E2.P$_i$ state), 9GSY (PMCA-alone E2P state) and 9GSI (PMCA-alone E1-Ca-ATP state). The atomic coordinates of the mouse NPTN Ig1 domain encompassing residues 29–148 determined by X-ray crystallography were deposited in the PDB under the ID 9GTI.

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

**Acknowledgements** We thank O. Hofnagel for electron microscopy technical support; and R. Gasper-Schönenbrücher for collecting data at the synchroton and support in X-ray crystallography. This work was supported by funds from the Max Planck Society (to S.R.), the German Research Foundation (TRR 152, project ID 239283807 to S.R. and B.F.; and CRC 1453, project ID 431984000, Fa 332/17-1 and 19-1 to B.F.). W.O. was supported by a postdoctoral fellowship from the Alexander von Humboldt foundation.

**Author contributions** S.R. and B.F. designed and supervised the study. U.S. prepared all protein complexes. C.E.C. and G.Z. carried out the electrophysiology experiments. D.V. and D.P. collected the cryo-EM data. D.V. processed the cryo-EM data, built the protein models for all datasets and solved the crystal structure of the NPTN N-terminal domain. D.V., U.S., W.O. and O.S. analysed the data. B.F. and S.R. evaluated the data. W.O., O.S., B.F. and S.R. prepared the figures. D.V. prepared Supplementary Figs. 1–5. O.S., B.F. and S.R. wrote the manuscript with critical input from all authors.

**Funding** Open access funding provided by Max Planck Society.

**Competing interests** The authors declare no competing interests.

**Additional information**
**Correspondence and requests for materials** should be addressed to Bernd Fakler or Stefan Raunser.

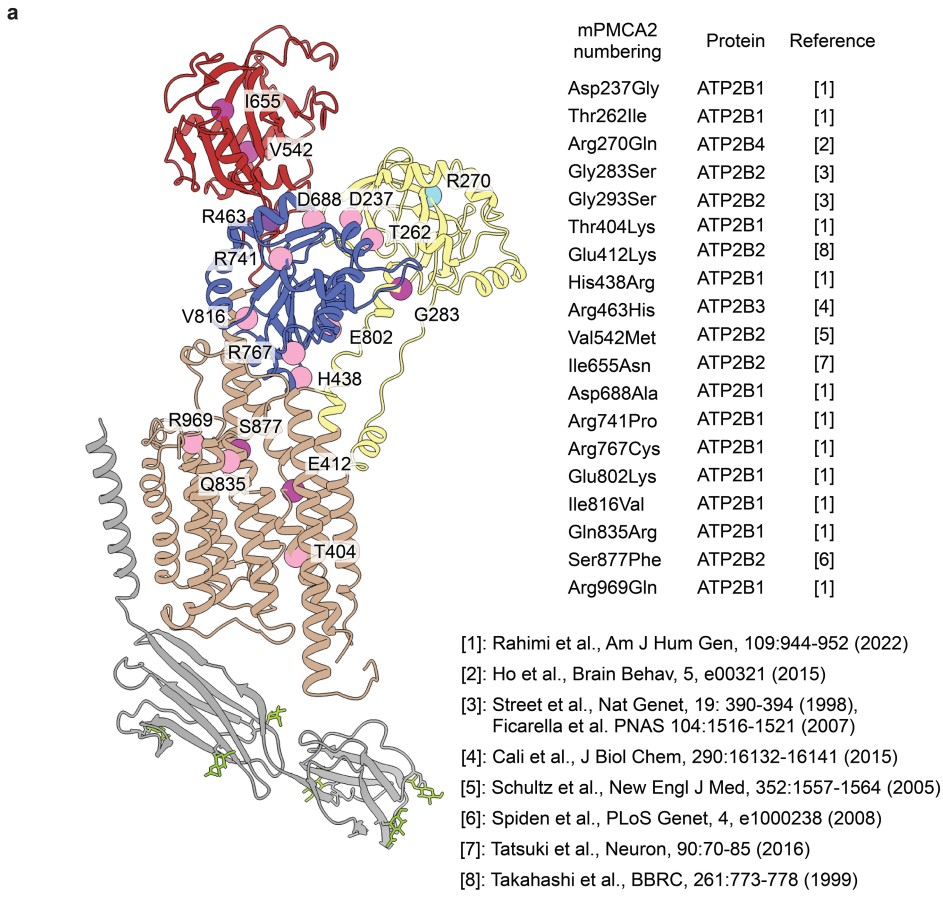

**a**

| mPMCA2 numbering | Protein | Reference |
|---|---|---|
| Asp237Gly | ATP2B1 | [1] |
| Thr262Ile | ATP2B1 | [1] |
| Arg270Gln | ATP2B4 | [2] |
| Gly283Ser | ATP2B2 | [3] |
| Gly293Ser | ATP2B2 | [3] |
| Thr404Lys | ATP2B1 | [1] |
| Glu412Lys | ATP2B2 | [8] |
| His438Arg | ATP2B1 | [1] |
| Arg463His | ATP2B3 | [4] |
| Val542Met | ATP2B2 | [5] |
| Ile655Asn | ATP2B2 | [7] |
| Asp688Ala | ATP2B1 | [1] |
| Arg741Pro | ATP2B1 | [1] |
| Arg767Cys | ATP2B1 | [1] |
| Glu802Lys | ATP2B1 | [1] |
| Ile816Val | ATP2B1 | [1] |
| Gln835Arg | ATP2B1 | [1] |
| Ser877Phe | ATP2B2 | [6] |
| Arg969Gln | ATP2B1 | [1] |

[1]: Rahimi et al., Am J Hum Gen, 109:944-952 (2022)

[2]: Ho et al., Brain Behav, 5, e00321 (2015)

[3]: Street et al., Nat Genet, 19: 390-394 (1998),
Ficarella et al. PNAS 104:1516-1521 (2007)

[4]: Cali et al., J Biol Chem, 290:16132-16141 (2015)

[5]: Schultz et al., New Engl J Med, 352:1557-1564 (2005)

[6]: Spiden et al., PLoS Genet, 4, e1000238 (2008)

[7]: Tatsuki et al., Neuron, 90:70-85 (2016)

[8]: Takahashi et al., BBRC, 261:773-778 (1999)

**b**

**Extended Data Fig. 1 | Disease-causing mutations of plasma membrane Ca²⁺-ATPases. a**, Structure of PMCA2-NPTN in the E2.P$_i$ state (left panel) with verified disease-causing mutations of PMCA1 highlighted in light pink, of PMCA2 highlighted in magenta, of PMCA3 highlighted in dark purple, and of PMCA4 highlighted in cyan. Right panel: Table of disease mutants in mouse PMCA2 numbering with respective references. **b**, Schematic representation of the Post-Albers cycle of PMCA2-NPTN. The states for which the cryo-EM structures have been determined in this study are highlighted in green.

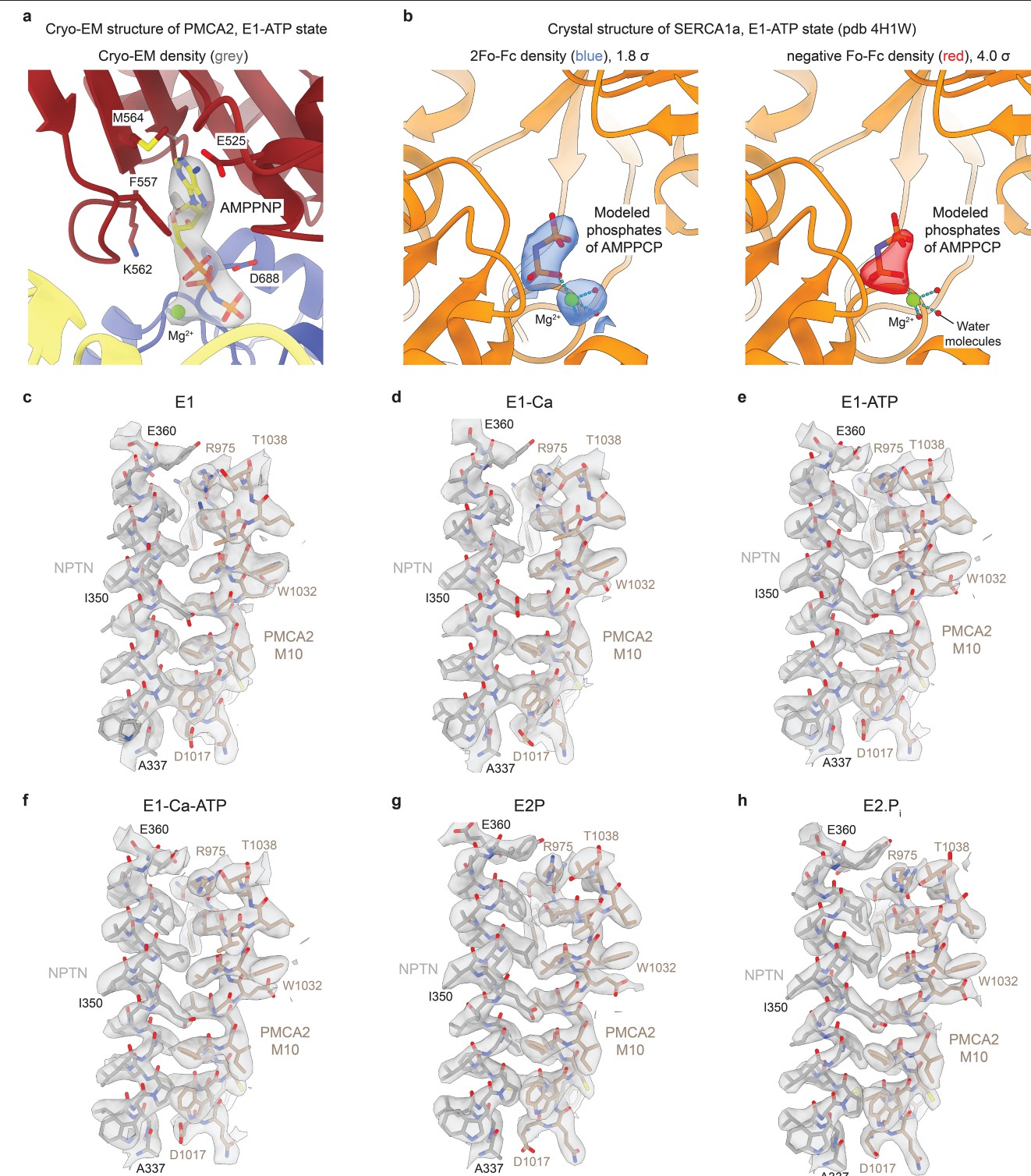

**Extended Data Fig. 2 | PMCA2 binds the nucleotide in the E1-ATP state more robustly than SERCA1a. a**, Depiction of the PMCA2 nucleotide binding site in the E1-ATP state. The bound AMPPNP molecule is shown, with the corresponding cryo-EM density in grey. **b**, Depiction of the nucleotide-binding pocket of the SERCA1a E1-ATP state crystal structure (PDB 4H1W), shown with corresponding 2Fo-Fc (left panel) and Fo-Fc (right panel) electron densities. In this structure, only the two terminal phosphates of AMPPCP were modelled. The strong negative Fo-Fc density around the two phosphates suggests that their occupancy is low. The structure and electron density of the SERCA1a E1-ATP state were retrieved from the PDB-REDO databank. **c-h**, Overlay of cryo-EM map and protein model for the PMCA–NPTN interaction region in structures of the complex in the E1 (**c**), E1-Ca (**d**), E1-ATP (**e**), E1-Ca-ATP (**f**), E2P (**g**) and E2.P$_i$ (**h**) states. For each state, NPTN residues 337–360 and PMCA2 residues 975–978 and 1017–1038 are depicted. The cryo-EM maps are shown semi-transparently and colored grey. There are no substantial changes in the PMCA–NPTN interaction between the different states of the transport cycle.

**a**

PMCA binding determinants (green: $Ca^{2+}$, purple: PtdIns(4,5)P$_2$, red: NPTN)

TM3

```
PMCA1  LQGKLTKLAVQIGKAGLLMSAITVIILVLYFVIDTFWVQKRPWLAE   407
PMCA2  LQGKLTKLAVQIGKAGLVMSAITVIILVLYFTVDTFVVNKKPWLTE   386
PMCA3  LQGKLTKLAVQIGKAGLVMSAITVIILVLYFVIETFVVDGRVWLAE   405
PMCA4  LQGKLTRLAVQIGKAGLIMSVLTVVILILYFVVDNFVIQRREWLPE   398
```

TM4

```
PMCA1  CTPIYIQYFVKFFIIGVTVLVVAVPEGLPLAVTISLAYSVKKMMKD   453
PMCA2  CTPVYVQYFVKFFIIGVTVLVVAVPEGLPLAVTISLAYSVKKMMKD   432
PMCA3  CTPVYVQYFVKFFIIGVTVLVVAVPEGLPLAVTISLAYSVKKMMKD   451
PMCA4  CTPVYIQYFVKFFIIGVTVLVVAVPEGLPLAVTISLAYSVKKMMKD   444
```

TM5                                              TM6

```
PMCA1  ISKFLQFQLTVNVVAVIVAFTGACITQDSPLKAVQMLWVNLIMDTL   897
PMCA2  ISKFLQFQLTVNVVAVIVAFTGACITQDSPLKAVQMLWVNLIMDTF   875
PMCA3  ISKFLQFQLTVNVVAVIVAFTGACITQDSPLKAVQMLWVNLIMDTF   894
PMCA4  ISKFLQFQLTVNVVAVIVAFTGACITQDSPLKAVQMLWVNLIMDTF   886
```

TM7

```
PMCA1  ASLALATEPPTESLLLRKPYGRNKPLISRTMMKNILGHAFYQLVVV   943
PMCA2  ASLALATEPPTETLLLRKPYGRNKPLISRTMMKNILGHAVYQLTLI   921
PMCA3  ASLALATEPPTESLLLRKPYGRDKPLISRTMMKNILGHAVYQLTII   940
PMCA4  ASLALATEPPTESLLRRRPYGRNKPLISRTMMKNILGHAVYQLLIV   932
```

TM8

```
PMCA1  FTLLFAGEKFFDIDSGRNAPLHAPPSEHYTIVFNTFVLMQLFNEIN   989
PMCA2  FTLLFVGEKMFQIDSGRNAPLHSPPSEHYTIIFNTFVMMQLFNEIN   967
PMCA3  FTLLFVGELFFDIDSGRNAPLHSPPSEHYTIIFNTFVMMQLFNEIN   986
PMCA4  FLLVFAGDTLFDIDSGRKAPLNSPPSQHYTIVFNTFVLMQLFNEIN   978
```

TM9

```
PMCA1  ARKIHGERNVFEGIFNNAIFCTIVLGTFVVQIIIVQFGGKPFSCSE   1035
PMCA2  ARKIHGERNVFDGIFRNPIFCTIVLGTFAIQIVIVQFGGKPFSCSP   1013
PMCA3  ARKIHGERNVFDGIFSNPIFCTIVLGTFGIQIVIVQFGGKPFSCSP   1032
PMCA4  ARKIHGEKNVFAGVYRNIIFCTVVLGTFFCQIMIVELGGKPFSCTS   1024
```

TM10

```
PMCA1  LSIEQWLWSIFLGMGTLLWGQLISTIPTSRLKFLKEAGHGTQKEEI   1081
PMCA2  LQLDQWMWCIFIGLGELVWGQVIATIPTSRLKFLKEAGRLTQKEEI   1059
PMCA3  LSTEQWLWCLFVGVGELVWGQVIATIPTSQLKCLKEAGHGPGKDEM   1078
PMCA4  LTMEQWMWCLFIGIGELLWGQVISAIPTKSLKFLKEAGHGSDKEDI   1070
```

**b**

PMCA interaction interface

```
EMBIGIN       VPKAHGKKKSLIAYVGDSTVLKCVCQDCL--PLNWTWYMGNETAQ-   202
BASIGIN       PPRIKVGKKSEHSSEGELAKLVCKSDASYPPITDWFWFKTSDTGEE   265
NEUROPLASTIN  APDITGHKRSENKNEGQDAMMYCKSVG--YPHPEWIWRKKENGV--   277
```

Ig-like 3

```
EMBIGIN       --VPIDAHSNEKYIINGSHANETRLKIKHLLEEDGGSYWCRATFQL   246
BASIGIN       EAITNSTEANGKYVVVSTPEKSQLTISNLDVNVDPGTYVCNATNAQ   311
NEUROPLASTIN  --FEEISNSSGRFFITNKENYTELSIVNLQITEDPGEYECNATNSI   321
```

TM

```
EMBIGIN       GESEEQNELVVLSFLVPLKPFLAILAEVILLVAIILLCEVYTHKKK   292
BASIGIN       GTTRETISLRVRSRMAALWPFLGIVAEVLVLVTIIFIYEKRR---K   354
NEUROPLASTIN  GSASVSTVLRVRSHLAPLWPFLGILAEIIILVVIIVVYEKRK---R   364
```

**Extended Data Fig. 3 | Sequence conservation of crucial PMCA interfaces.**
**a**, Conservation of residues involved in binding of $Ca^{2+}$ (green), PtdIns(4,5)P$_2$ (purple) and NPTN (red) in mouse PMCA1-4. **b**, Sequence alignment of the PMCA interaction interface with the mouse basigin family of proteins. Conservation in all three proteins is highlighted by orange boxes and red boxes indicate conservation only between basigin and NPTN.

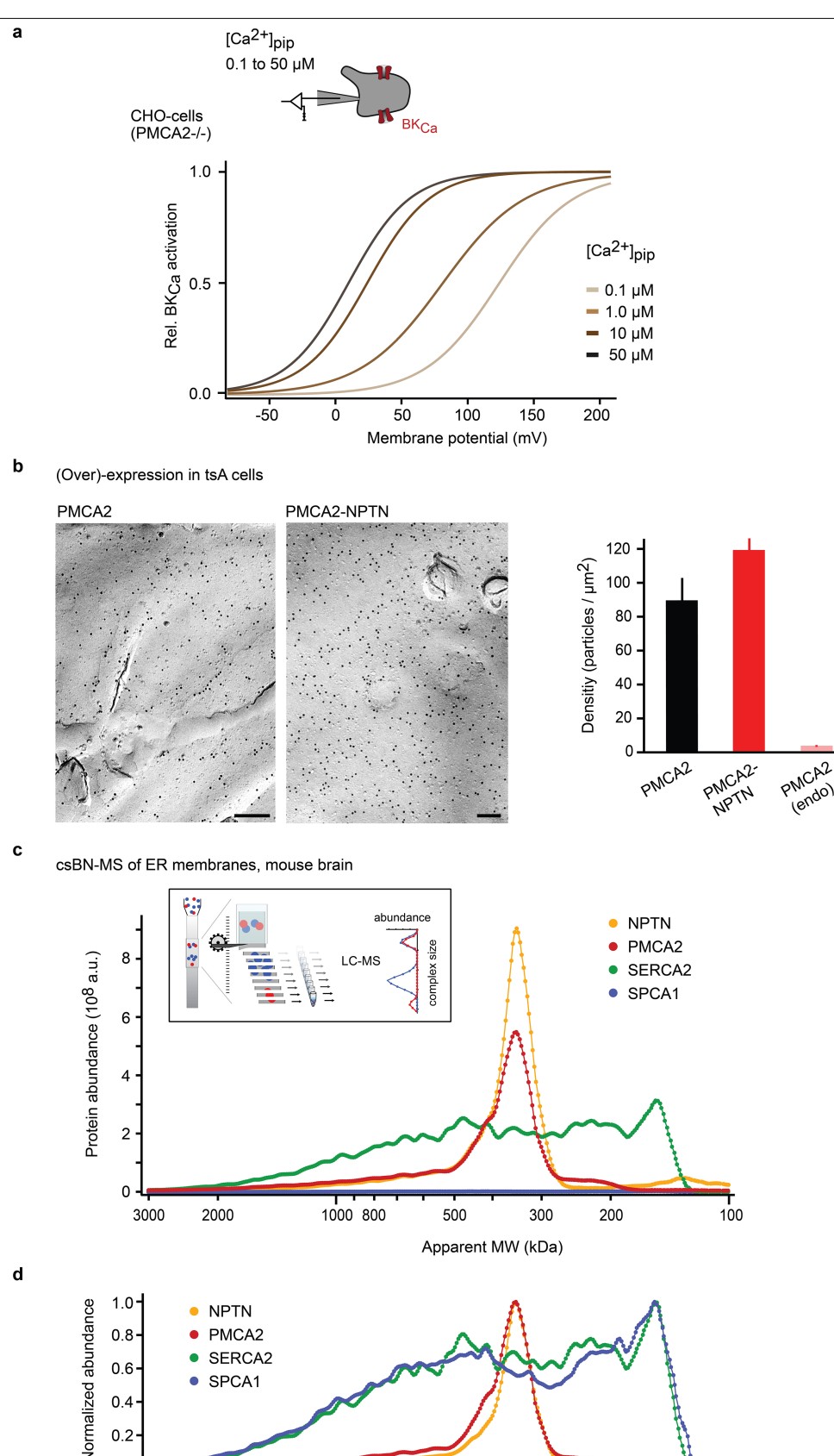

**Extended Data Fig. 4** | See next page for caption.

**Extended Data Fig. 4 | BK$_{Ca}$ as a reporter for intracellular Ca$^{2+}$ concentration and EM-derived surface density of PMCA2. a**, Dependence of BK$_{Ca}$ channel activation on intracellular Ca$^{2+}$ concentration at the plasma membrane used for monitoring the Ca$^{2+}$ transport of PMCA2-NPTN. Steady-state activation curves of BK$_{Ca}$ channels obtained by fitting a Boltzmann-function to the K$^{+}$ current-recordings performed with whole-cell patch-clamping in CHO cells with the indicated Ca$^{2+}$ concentrations in the patch-pipette (inset). For eliminating effects of CHO-endogenous PMCA2, this protein was deleted by Crispr/Cas technology (Methods). **b**, Efficient protein overexpression overcomes the requirement for NPTN-mediated trafficking of PMCA2 to the cell surface. Left panel, electron micrographs showing distribution of immunogold particles for PMCA2 in freeze-fracture replicas of tsA cells expressing PMCA2 either alone (left) or together with NPTN (right). Scale bars are 200 nm. Right panel, Bar graph illustrating the density of PMCA2 determined under the indicated conditions. Data are mean ± SD of 10 cells for each condition. Note the similar densities of PMCA2 observed in over-expressions largely exceeding the protein level endogenous to tsA cells. **c, d**, Analysis by csBN-MS of Ca$^{2+}$-ATPases in ER-enriched fractions of adult mouse brain reveals their distinct molecular appearances. Absolute (**c**) and normalized (**d**) abundance-mass profiles obtained for the indicated proteins by csBN-MS analysis of ComplexioLyte-47 solubilized ER-enriched membrane fractions from mouse brain. Inset: Scheme of the csBN-MS approach used (also see Methods). Note the close co-migration of PMCA2 and NPTN on the native gel due to their tight co-assembly and the large peak corresponding to the abundant bi-molecular PMCA2-NPTN complex. SERCA2 and SPCA1 exhibit an entirely different appearance: the maximal peak at about 150 kDa corresponds to the ATPase subunits, while high-molecular mass peaks reflect their co-assembly with other ER proteins.

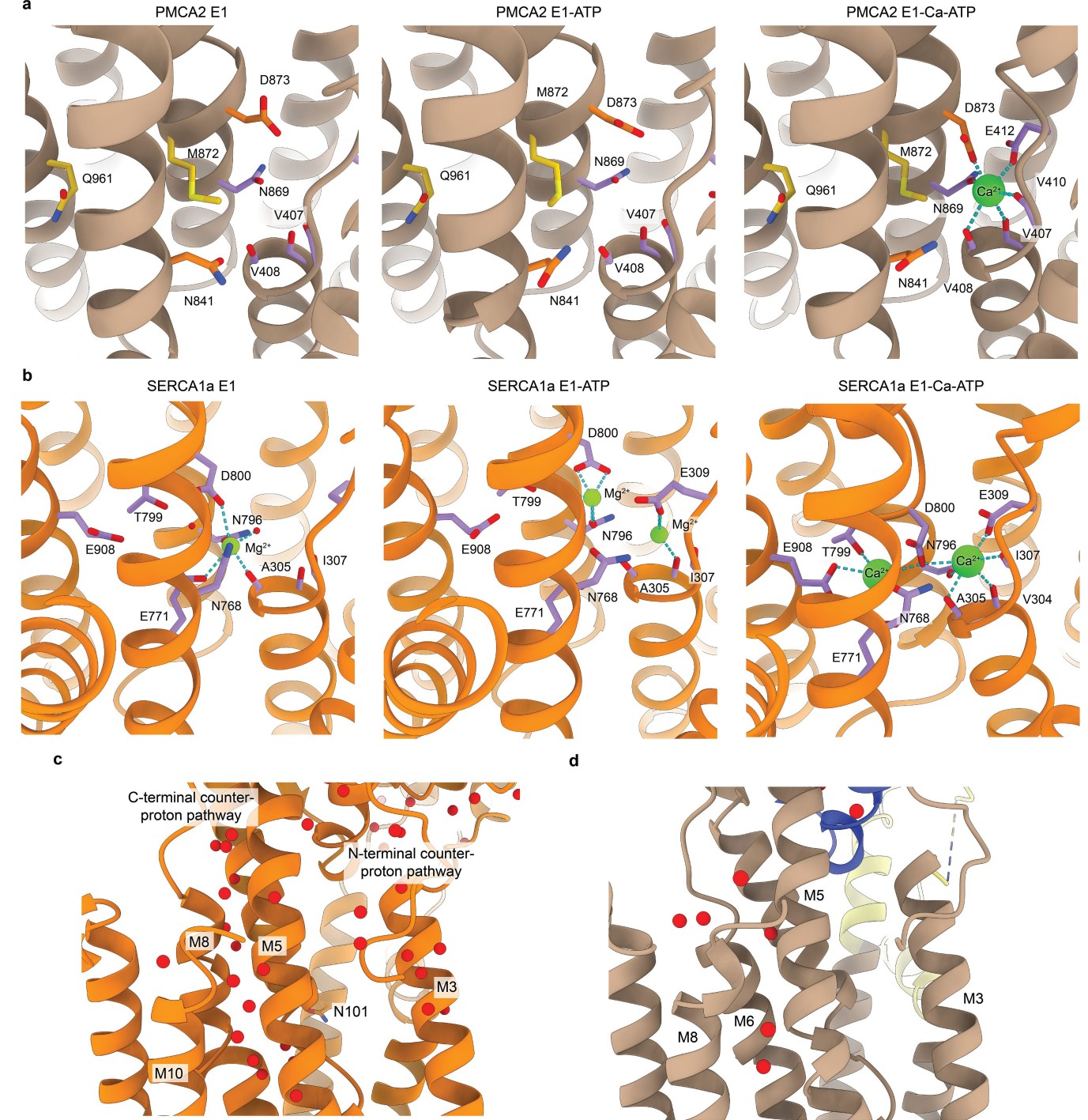

**Extended Data Fig. 5 | Pre-positioning of Ca²⁺-binding residues for ion binding in PMCA2 and counterion release in SERCA1a. a**, Cartoon representation of the ion binding residues of PMCA2 in the E1, E1-ATP, and E1-Ca-ATP states, demonstrating that nucleotide binding rearranges residues into a conformation similar to the succeeding E1-Ca-ATP state. The PMCA2 residues are depicted in three different colors: the purple residues participate in Ca²⁺ binding in PMCA, equivalent to SERCA1a Ca²⁺-binding site 2. The orange residues are unchanged from the SERCA1a Ca²⁺-binding site 1, but are not involved in Ca²⁺-binding in PMCA. The yellow residues differ from those of the

SERCA1a Ca²⁺-binding site 1 and are not used in PMCA for Ca²⁺ binding. **b**, Views corresponding to those of (**a**) for SERCA1a. Residues involved in ion binding are shown in light purple. **c**, The two SERCA1a counterion pathways (N- and C-terminal) are lined by water molecules in the high resolution E2.Pᵢ-Tg structure (PDB ID: 3N5K). The water wire stabilizing N101 of SERCA1a is marked. **d**, Cartoon representation of PMCA2-NPTN in the E2.Pᵢ state, in the same orientation as SERCA1a in panel **c**. The ordered waters in this structure suggest that PMCA2 employs a single counterion pathway that is similar to the C-terminal pathway of SERCA1a.

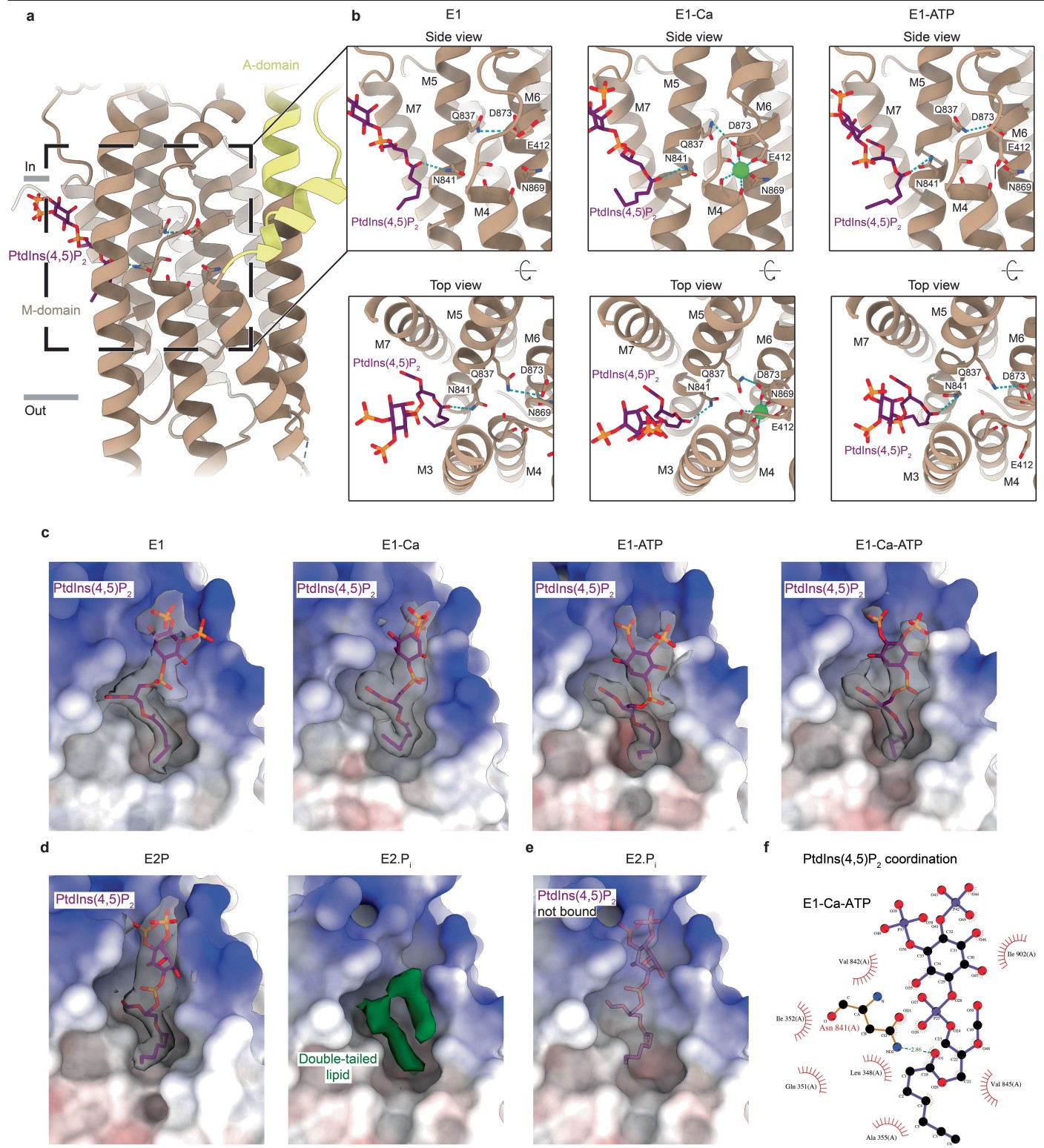

**Extended Data Fig. 6** | See next page for caption.

**Extended Data Fig. 6 | Binding and dynamics of PtdIns(4,5)P$_2$ to PMCA2.**
**a**, Structure of the transmembrane region of PMCA2 in the E1 state. The PtdIns(4,5)P$_2$ and Ca$^{2+}$-binding sites are shown. **b**, Zooms of the connection between the PtdIns(4,5)P$_2$ and Ca$^{2+}$-binding sites in structures of PMCA2-NPTN in the E1 (left), E1-Ca (middle) and E1-ATP (right) states. Residues involved in Ca$^{2+}$ binding are shown as sticks, as well as residues N841 and Q837 that mediate the interaction of PtdIns(4,5)P$_2$ with the Ca$^{2+}$-binding site. The top panels depict the protein models parallel to the membrane as side view, while the bottom panels show the models from the cytoplasmic side of the membrane as top view. In the side views, membrane helices M1 (residues 100 – 120) and M3 (residues 339–377) are hidden for visualization purposes. See also Supplementary Videos 4 and 5. **c**, The PtdIns(4,5)P$_2$ binding site of PMCA2-NPTN in the E1, E1-Ca, E1-ATP and E1-Ca-ATP states. PtdIns(4,5)P$_2$ is shown in stick representation. PMCA2 is shown as surface. The PtdIns(4,5)P$_2$ head group displays a dynamic mode of binding throughout the E1 state structures, while the tail is relatively rigid. **d**, The PtdIns(4,5)P$_2$ binding site of PMCA2-NPTN in the E2P and E2.P$_i$ states. PtdIns(4,5)P$_2$ is shown in stick representation. PMCA2 is shown as surface. PtdIns(4,5)P$_2$ is absent in the E2.P$_i$ state structure, with an unmodelled double-tailed lipid (green density) taking its place. **e**, Fit of the PtdIns(4,5)P$_2$ in the PMCA2-NPTN E2.P$_i$ state structure. PtdIns(4,5)P$_2$ is shown semi-transparently to emphasize that it is not bound. The fit suggests that the binding of PtdIns(4,5)P$_2$ is no longer favored in the E2.P$_i$ state because of changes in the pocket. In panels **c-e**, the PMCA surface is colored by electrostatic Coulomb potential ranging from −10 kcal (mol e)$^{-1}$ (red) to +10 kcal (mol e)$^{-1}$ (blue). **f**, 2D protein-ligand interaction diagram illustrating the coordination of PtdIns(4,5)P$_2$ in the E1-Ca-ATP state.

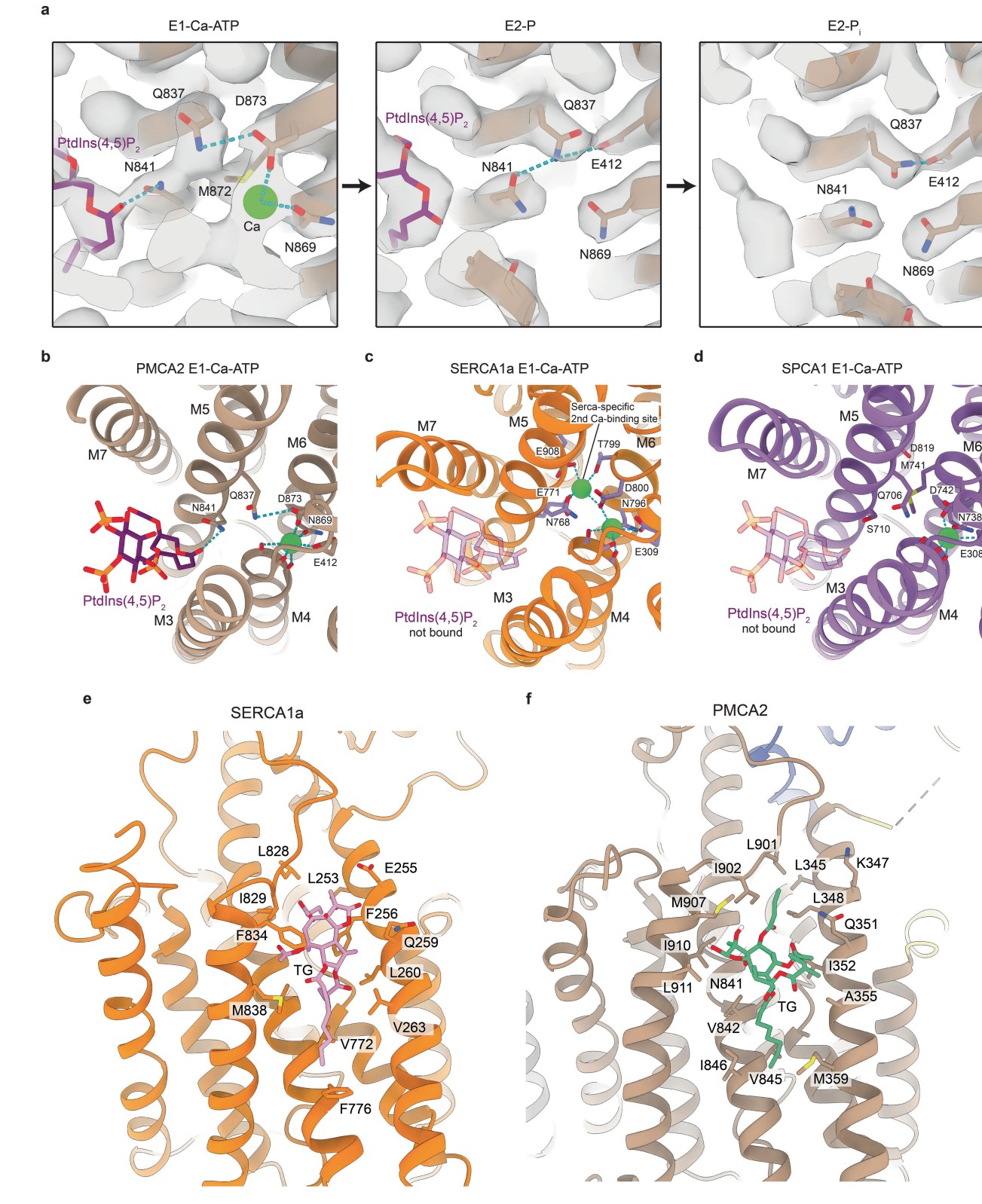

**Extended Data Fig. 7** | See next page for caption.

**Extended Data Fig. 7 | Comparison of the PtdIns(4,5)P$_2$-binding pocket in PMCA2 with the equivalent site in other Ca$^{2+}$-ATPases. a**, Zooms of the connection between the PtdIns(4,5)P$_2$ and Ca$^{2+}$-binding sites in structures of PMCA2-NPTN in the E1-Ca-ATP (left), E2P (middle) and E2.P$_i$ (right) states with corresponding cryo-EM densities. Residues involved in Ca$^{2+}$ binding are shown as sticks, as well as residues N841 and Q837 that mediate the interaction of PtdIns(4,5)P$_2$ with the Ca$^{2+}$-binding site. PtdIns(4,5)P$_2$ is not bound in the E2.P$_i$ state, but the molecule is shown semi-transparently to emphasize its binding site. The cryo-EM density is shown in transparent grey. The panels show a zoomed and clipped version of the side views depicted in Fig. 5d, but are shown in the same orientation. **b-d**, Structures of the E1-Ca-ATP states of PMCA2-NPTN (**b**, this study), SERCA1a (**c**, PDB ID 3BA6) and SPCA1 (**d**, PDB ID 7YAH)

shown orthogonally to the membrane from the cytoplasmic side. In PMCA2, residue N841 binds to PtdIns(4,5)P$_2$ and is important for connecting the PtdIns(4,5)P$_2$ binding site to the Ca$^{2+}$-binding site. The equivalent residue in SERCA1a (N768) is involved in ion binding, while the equivalent residue in SPCA1 (S710) is too short to maintain a potential interaction with PtdIns(4,5)P$_2$. Accordingly, PtdIns(4,5)P$_2$-mediated control of Ca$^{2+}$-transport only occurs in PMCAs, but not in SERCA or SPCA. **e,f**, Binding of thapsigargin to SERCA1a compared to PMCA2. The thapsigargin site in SERCA1a (**c**) in the E2.P$_i$ state (PDB ID 5XA9), with residues involved in thapsigargin binding shown in light red. The PtdIns(4,5)P$_2$ binding site of PMCA2 (**d**) in the E2P state with docked thapsigargin from Fig. 6a is shown for comparison.

**Extended Data Table 1 | Cryo-EM data collection, refinement and validation statistics for various states of the PMCA2-NPTN complex and PMCA2**

| | E1-Ca (EMDB-51560) (PDB 9GTB) | E1-Ca-AMPPNP (EMDB-51548) (PDB 9GSH) | E2 (EMDB-51544) (PDB 9GSD) | E2-Pi (EMDB-51547) (PDB 9GSG) | E1-AMPPNP (EMDB-51546) (PDB 9GSF) | E1 (EMDB-51545) (PDB 9GSE) | PMCA2 E1-Ca-AMPPNP (EMDB-51549) (PDB 9GSI) | PMCA2 E2 (EMDB-51558) (PDB 9GSY) |
|---|---|---|---|---|---|---|---|---|
| **Data collection and processing** | | | | | | | | |
| Magnification | 105,000 | 105,000 | 105,000 | 105,000 | 105,000 | 105,000 | 105,000 | 105,000 |
| Voltage (kV) | 300 | 300 | 300 | 300 | 300 | 300 | 300 | 300 |
| Electron exposure (e–/Å$^2$) | 67.6 | 60.0 | 60.5 | 59.5 | 59.0 | 59.0 | 56.7 | 57.0 |
| Defocus range (μm) | -1.1to-2.5 | -1.2to-2.4 | -1.2to-2.4 | -1.2to-2.4 | -1.2to-2.4 | -1.2to-2.4 | -1.2to-2.4 | -1.2to-2.4 |
| Pixel size (Å) | 0.91 | 0.91 | 0.91 | 0.91 | 0.91 | 0.91 | 0.91 | 0.91 |
| Symmetry imposed | C1 | C1 | C1 | C1 | C1 | C1 | C1 | C1 |
| Initial particle images (no.) | 3,337,412 | 1,655,342 | 1,318,216 | 1,254,462 | 1,350,753 | 1,369,679 | 1,904,953 | 1,187,705 |
| Final particle images (no.) | 201,573 | 303,557 | 323,585 | 299,645 | 185,466 | 299,645 | 300,272 | 437,154 |
| Map resolution (Å) FSC threshold | 3.50 | 3.13 | 3.04 | 2.80 | 3.35 | 3.64 | 3.39 | 3.52 |
| Map resolution range (Å) | 3.1-7.2 | 2.7-6.5 | 2.6-6.3 | 2.3-4.5 | 2.8-6.5 | 3.1-7.0 | 3.0-9.5 | 3.1-5.9 |
| | | | | | | | | |
| **Refinement** | | | | | | | | |
| Initial model used (PDB code) | 9GSG | 9GSG | 9GSG | 6A69 | 9GSG | 9GSG | 9GSG | 9GSG |
| Model resolution (Å) FSC threshold | 3.7 | 3.2 | 3.2 | 3.0 | 3.5 | 3.8 | 3.6 | 3.6 |
| Map sharpening B factor (Å$^2$) | -146.2 | -128.9 | -125.1 | -113.08 | -133.2 | -191.4 | -161.4 | -181.1 |
| Model composition | | | | | | | | |
| Non-hydrogen atoms | 10,064 | 9,177 | 9,182 | 9,093 | 9,195 | 9,155 | 7,316 | 7,353 |
| Protein residues | 1,271 | 1,149 | 1,161 | 1,153 | 1,156 | 1,156 | 933 | 941 |
| Ligands | 8 | 10 | 9 | 8 | 9 | 7 | 4 | 3 |
| Waters | - | - | 2 | 10 | - | - | - | - |
| B factors (Å$^2$) | | | | | | | | |
| Protein | 108.08 | 95.44 | 114.72 | 118.28 | 119.03 | 120.79 | 93.29 | 88.34 |
| Ligand | 117.53 | 112.14 | 92.40 | 206.36 | 153.41 | 143.35 | 77.41 | 66.16 |
| Waters | | | 55.22 | 43.07 | | | | |
| R.m.s. deviations | | | | | | | | |
| Bond lengths (Å) | 0.006 | 0.004 | 0.003 | 0.005 | 0.004 | 0.004 | 0.003 | 0.003 |
| Bond angles (°) | 0.754 | 0.699 | 0.592 | 0.759 | 0.665 | 0.738 | 0.651 | 0.513 |
| Validation | | | | | | | | |
| MolProbity score | 2.17 | 1.89 | 1.55 | 1.72 | 1.82 | 1.89 | 1.66 | 1.53 |
| Clashscore | 10.32 | 7.34 | 4.86 | 6.76 | 6.20 | 8.87 | 6.07 | 3.76 |
| Poor rotamers (%) | 2.08 | 1.5 | 0.89 | 0.0 | 1.68 | 1.49 | 0.62 | 1.10 |
| EM-ringer score | 1.23 | 2.13 | 2.46 | 3.02 | 1.95 | 1.18 | 2.04 | 1.75 |
| Ramachandran plot | | | | | | | | |
| Favored (%) | 94.13 | 94.82 | 95.74 | 95.00 | 95.64 | 95.91 | 95.24 | 95.18 |
| Allowed (%) | 5.55 | 5.09 | 4.17 | 4.91 | 4.27 | 4.09 | 4.54 | 4.82 |
| Disallowed (%) | 0.32 | 0.09 | 0.09 | 0.09 | 0.09 | 0.0 | 0.22 | 0.0 |

**Extended Data Table 2 | X-ray diffraction data collection and refinement statistics (molecular replacement)**

| | NPTN (N-domain) |
|---|---|
| **Data collection** | |
| Space group | $P3_221$ |
| Cell dimensions | |
| $a, b, c$ (Å) | 85.91, 85.91, 163.285 |
| $\alpha, \beta, \gamma$ (°) | 90, 90, 120 |
| Resolution (Å) | 43.93-2.031 (2.103-2.031) |
| $R_{sym}$ or $R_{merge}$ | 0.2083 (2.797) |
| $I / \sigma I$ | 12.07 (1.14) |
| Completeness (%) | 99.74 (97.83) |
| Redundancy | 20.2 (20.5) |
| | |
| **Refinement** | |
| Resolution (Å) | 43.93-2.031 |
| No. reflections | 45685 (4410) |
| $R_{work} / R_{free}$ | 0.216/0.249 |
| No. atoms | |
| Protein | 3747 |
| Ligand/ion | 120 |
| Water | 189 |
| $B$-factors | |
| Protein | 41.72 |
| Ligand/ion | 56.10 |
| Water | 46.59 |
| R.m.s. deviations | |
| Bond lengths (Å) | 0.008 |
| Bond angles (°) | 0.87 |

*Number of xtals: 1, *Values in parentheses are for highest-resolution shell.

# Reporting Summary

## Statistics

For all statistical analyses, confirm that the following items are present in the figure legend, table legend, main text, or Methods section.

| n/a | Confirmed | |
|---|---|---|
| ☒ | ☐ | The exact sample size (*n*) for each experimental group/condition, given as a discrete number and unit of measurement |
| ☒ | ☐ | A statement on whether measurements were taken from distinct samples or whether the same sample was measured repeatedly |
| ☒ | ☐ | The statistical test(s) used AND whether they are one- or two-sided<br>*Only common tests should be described solely by name; describe more complex techniques in the Methods section.* |
| ☒ | ☐ | A description of all covariates tested |
| ☒ | ☐ | A description of any assumptions or corrections, such as tests of normality and adjustment for multiple comparisons |
| ☒ | ☐ | A full description of the statistical parameters including central tendency (e.g. means) or other basic estimates (e.g. regression coefficient) AND variation (e.g. standard deviation) or associated estimates of uncertainty (e.g. confidence intervals) |
| ☒ | ☐ | For null hypothesis testing, the test statistic (e.g. *F*, *t*, *r*) with confidence intervals, effect sizes, degrees of freedom and *P* value noted<br>*Give P values as exact values whenever suitable.* |
| ☒ | ☐ | For Bayesian analysis, information on the choice of priors and Markov chain Monte Carlo settings |
| ☒ | ☐ | For hierarchical and complex designs, identification of the appropriate level for tests and full reporting of outcomes |
| ☒ | ☐ | Estimates of effect sizes (e.g. Cohen's *d*, Pearson's *r*), indicating how they were calculated |

*Our web collection on statistics for biologists contains articles on many of the points above.*

## Software and code

Policy information about availability of computer code

| | |
|---|---|
| Data collection | Cryo-EM data was collected using the commercially available software EPU version 2.8 (ThermoFisher Scientific). |
| Data analysis | Data collection was monitored live using TranSPHIRE. Intial motion correction and dose weighting was done with  MotionCor2 v.1.3.0 and CTF estimation  with CTFFIND 4.1.131. SPHIRE software package version 1.4 was used for Particle extraction and 2D classification.  Particles were picked automatically with crYOLO version 1.8. 3D classification and refinement was performed with Relion v 3.1 and CryoSPARC v 4.0. Protein model building was  carried out in Coot v1.9 and further refined by Phenix v1.18.2. Figures were prepared in Chimera v 1.15 and Chimera X v 1.7.1.  X-ray crystallography data processing was carried out in XDS (VERSION Jan 10, 2022  BUILT=20220220) and Phenix v1.18.2 followed by model building and refinement in Coot v1.9 and Phenix v1.18.2, respectively. The residue displacement plot shown in Fig. 2c was generated using a custom script obtained from: https://github.com/schaefer-jh/motionviz.<br><br>msconvert v3.0.11098 was used for processing of primary MS data, MaxQuant 1.6.17 for calibration and quantification of MS data, Mascot 2.7 for database search (UniProtKB/SwissProt release 20181205), and Igor Pro 9 (Wavemetrics) for data fitting and figure preparation. |

For manuscripts utilizing custom algorithms or software that are central to the research but not yet described in published literature, software must be made available to editors and reviewers. We strongly encourage code deposition in a community repository (e.g. GitHub). See the Nature Portfolio guidelines for submitting code & software for further information.

## Data

Policy information about availability of data

 All manuscripts must include a data availability statement. This statement should provide the following information, where applicable:

- Accession codes, unique identifiers, or web links for publicly available datasets
- A description of any restrictions on data availability
- For clinical datasets or third party data, please ensure that the statement adheres to our policy

The cryo-EM maps were deposited in the EMBD with accession ID (dataset in brackets) : EMD-51545 (PMCA-NPTN  E1 state), EMD-51546 (PMCA-NPTN  E1-ATP state), EMD-51560  (PMCA-NPTN E1-Ca state), EMD-51625 (PMCA-NPTN E1-Ca state used to fit in NPTN Ig1), EMD-51548 (PMCA-NPTN E1-Ca-ATP state), EMD-51544 (PMCA-NPTN E2P state),EMD-51547 (PMCA-NPTN E2.Pi state), EMD-51558 (PMCA alone E2P state) and  EMD-51549 (PMCA-alone E1-Ca-ATP  state). The atomic  coordinates were deposited in the PDB databank under the  accession ID (dataset  in brackets) : PDB ID 9GSE (PMCA-NPTN  E1 state),  PDB ID 9GSF (PMCA-NPTN  E1-ATP state), PDB ID 9GTB  (PMCA-NPTN E1-Ca state), PDB ID 9GSH (PMCA-NPTN E1-Ca-ATP state), PDB ID 9GSD (PMCA-NPTN E2P state),PDB ID 9GSG (PMCA-NPTN E2.Pi state), PDB ID 9GSY (PMCA alone E2P state) and PDB ID 9GSI (PMCA-alone E1-Ca-ATP  state),

The atomic coordinates of the mouse NPTN Ig1 domain encompassing residue 29-148 determined by X-ray crystallography were deposited in the PDB databank under the PDB ID 9GTI.

## Research involving human participants, their data, or biological material

Policy information about studies with human participants or human data. See also policy information about sex, gender (identity/presentation), and sexual orientation and race, ethnicity and racism.

| | |
|---|---|
| Reporting on sex and gender | *Use the terms sex (biological attribute) and gender (shaped by social and cultural circumstances) carefully in order to avoid confusing both terms. Indicate if findings apply to only one sex or gender; describe whether sex and gender were considered in study design; whether sex and/or gender was determined based on self-reporting or assigned and methods used.*<br>*Provide in the source data disaggregated sex and gender data, where this information has been collected, and if consent has been obtained for sharing of individual-level data; provide overall numbers in this Reporting Summary.  Please state if this information has not been collected.*<br>*Report sex- and gender-based analyses where performed, justify reasons for lack of sex- and gender-based analysis.* |
| Reporting on race, ethnicity, or other socially relevant groupings | *Please specify the socially constructed or socially relevant categorization variable(s) used in your manuscript and explain why they were used. Please note that such variables should not be used as proxies for other socially constructed/relevant variables (for example, race or ethnicity should not be used as a proxy for socioeconomic status).*<br>*Provide clear definitions of the relevant terms used, how they were provided (by the participants/respondents, the researchers, or third parties), and the method(s) used to classify people into the different categories (e.g. self-report, census or administrative data, social media data, etc.)*<br>*Please provide details about how you controlled for confounding variables in your analyses.* |
| Population characteristics | *Describe the covariate-relevant population characteristics of the human research participants (e.g. age, genotypic information, past and current diagnosis and treatment categories). If you filled out the behavioural & social sciences study design questions and have nothing to add here, write "See above."* |
| Recruitment | *Describe how participants were recruited. Outline any potential self-selection bias or other biases that may be present and how these are likely to impact results.* |
| Ethics oversight | *Identify the organization(s) that approved the study protocol.* |

Note that full information on the approval of the study protocol must also be provided in the manuscript.

# Field-specific reporting

Please select the one below that is the best fit for your research. If you are not sure, read the appropriate sections before making your selection.

☒ Life sciences ☐ Behavioural & social sciences ☐ Ecological, evolutionary & environmental sciences

For a reference copy of the document with all sections, see nature.com/documents/nr-reporting-summary-flat.pdf

# Life sciences study design

All studies must disclose on these points even when the disclosure is negative.

| | |
|---|---|
| Sample size | Sample size for eight different cryo-EM datasets is given below<br>1. For PMCA-NPTN E1 state 6,676 movies were collected. 292,379 particles were used for final reconstruction.<br>2. For PMCA-NPTN E-Ca state 29,014 movies were collected. 105,000 best particles were used to resolve the neuroplastin terminal domain. 201,573 particles were used to resolve the cytosolic domain and also for final reconstruction.<br>3. For PMCA-NPTN E1-Ca-ATP  state 5268 movies were collected. 303,557 particles were used for final reconstruction.<br>4. For PMCA-NPTN E1-ATP  state 5788 movies were collected. 185,466 particles were used for final reconstruction. |

5. For PMCA-NPTN E2P state 6266 movies were collected. 325,585 particles were used for final reconstruction.
6. For PMCA-NPTN E2.Pi state 4611 movies were collected. 352,232 particles were used for final reconstruction.
7. For PMCA alone E1-Ca-ATP state 8076 movies were collected. 300,272 particles were used for final reconstruction.
8. For PMCA alone E2P state 13134 movies were collected. 473,154 particles were used for final reconstruction.

| | |
|---|---|
| Data exclusions | During cryo-EM dataset processing, false picks during particle picking were eliminated and further particles that do not contribute to high resolution features have been removed based on 2D and 3D classification which is a standard procedure. |
| Replication | All cryo-EM dataset were acquired once as it is unattainable to repeat the cryo-EM dataset collection and processing of the same sample from a time and cost perspective. |
| Randomization | During final cryo-EM dataset reconstruction particles were randomly split into two equal subsets for FSC calculation. |
| Blinding | Not applicable for this experiment |

# Reporting for specific materials, systems and methods

We require information from authors about some types of materials, experimental systems and methods used in many studies. Here, indicate whether each material, system or method listed is relevant to your study. If you are not sure if a list item applies to your research, read the appropriate section before selecting a response.

### Materials & experimental systems

| n/a | Involved in the study |
|---|---|
| ☒ | Antibodies |
| ☐ | ☒ Eukaryotic cell lines |
| ☒ | Palaeontology and archaeology |
| ☒ | Animals and other organisms |
| ☒ | Clinical data |
| ☒ | Dual use research of concern |
| ☒ | Plants |

### Methods

| n/a | Involved in the study |
|---|---|
| ☒ | ChIP-seq |
| ☒ | Flow cytometry |
| ☒ | MRI-based neuroimaging |

## Eukaryotic cell lines

Policy information about cell lines and Sex and Gender in Research

| | |
|---|---|
| Cell line source(s) | - tsa201 nptn/basi double knockout (described in Schmidt et al., Neuron, 96: 827-838 (2017), derived from tsa201, Sigma, Cat#96121229)<br>- CHO-K1 (Leibniz Institute DSMZ-German Collection of Microorganisms and Cell Culture, Cat#ACC110)<br>- CHO-K1 at2b2-4 -/- |
| Authentication | Cell lines were not authenticated |
| Mycoplasma contamination | There was no contamination |
| Commonly misidentified lines (See ICLAC register) | Commonly misidentified lines were not used in this study |

## Plants

| | |
|---|---|
| Seed stocks | *Report on the source of all seed stocks or other plant material used. If applicable, state the seed stock centre and catalogue number. If plant specimens were collected from the field, describe the collection location, date and sampling procedures.* |
| Novel plant genotypes | *Describe the methods by which all novel plant genotypes were produced. This includes those generated by transgenic approaches, gene editing, chemical/radiation-based mutagenesis and hybridization. For transgenic lines, describe the transformation method, the number of independent lines analyzed and the generation upon which experiments were performed. For gene-edited lines, describe the editor used, the endogenous sequence targeted for editing, the targeting guide RNA sequence (if applicable) and how the editor was applied.* |
| Authentication | *Describe any authentication procedures for each seed stock used or novel genotype generated. Describe any experiments used to assess the effect of a mutation and, where applicable, how potential secondary effects (e.g. second site T-DNA insertions, mosiacism, off-target gene editing) were examined.* |

