## [Peer Review File · Nature]

Molecular mechanism of ultrafast transport by plasma membrane Ca²⁺-ATPases

Corresponding Author: Professor Stefan Raunser

Version 0:

Reviewer comments:

Referee #1

(Remarks to the Author)

My comments regarding the manuscript "Molecular Mechanism..." by Vinayagam et al are confined solely to the use of BK channels expressed in COS cells to monitor the ability of WT or manipulated plasma membrane Ca²⁺-ATPase to reduce cytosolic Ca²⁺. Because BK channels are regulated by both Ca²⁺ and voltage, in whole cell recordings determination of the membrane potential at which BK channels are half activated provides a fairly accurate estimate of the actual cytosolic Ca²⁺ concentration. Based on results presented in the companion paper, the authors demonstrated that coexpression of PMCA-NPTN effectively and dramatically can reduce a nominal cytosolic Ca²⁺ concentration of 10 micromolar, down to near 100 nM. With this assay, the authors can then potentially manipulate PMCA2 function allowing results that may inform their structures.

Fig 3d clearly demonstrates that whether PMCA2 is expressed alone or with NPTN, the clearance of Ca²⁺ from the COS cells is similar.

Fig. 4 provides structures of two disease-associated mutant PMCA2 transporters and Fig. 4b nicely confirms that both mutants that are evaluated (E412K and S877F) are hindered in their ability to remove Ca²⁺, with E412K interfering with Ca²⁺ coordination and S877F altering counterion transport.

Figure 5 examines the requirement for PIP₂ in PMCA2 function, and the authors use two functional experiments to validate the impact of manipulations of PIP₂ on their BK-COS Ca²⁺ clearance assay. In Fig. 5a, using the Ciona voltage-sensitive phosphatase which rapidly reduces membrane concentrations of PIP₂, the authors show that with CiVSP stimulation produces a profound loss of ability to PMCA2 to clear Ca²⁺. My only concern here is that in the text the authors mention "this was done using co-expressed PIP₂-phosphatase CiVSP that effectively depletes PIP₂ after activation of its enzymatic activity (Methods)." Unfortunately, there appears to be no mention in the Methods or in the figure legends of how CiVSP was activated. In Fig. 5B, the authors examine functional impact of two mutations which alter PIP₂ interactions in their structures. Both of these mutations that disrupt PIP₂ binding result in reduced ability of the mutant PCMA2 proteins to clear Ca²⁺.

Finally, the authors explore the possibility that thapsigargin, an inhibitor of SERCA mediated Ca²⁺ uptake, may also interact with PMCA2. This is supported by docking computational tests, but apparently not structural data. The hypothesis that thapsigargin does interact with PMCA2 is supported by functional data in Fig. 6c that clearly shows that application of thapsigargin to cells expressing PMCA2 hinders the clearance of Ca²⁺ in a nicely concentration dependent fashion.

Overall, the functional tests are straight-forward and seem to nicely complement the structures.

Referee #2

(Remarks to the Author)

In this work, the authors report cryo-EM structures of plasma membrane Ca²⁺-ATPase (PMCA) complexed with its functional

partner NPTN in 6 different states and of PMCA alone in two different states. In all cryo-EM density maps but that of the E2Pi state, a density corresponding to PIP2, a modulator of PMCA, was observed. They also conducted functional assays based on the structural information by monitoring the PMCA-mediated Ca²⁺ extrusion using Ca²⁺-activated BK channels as reporters. The totally eight cryo-EM structures of PMCA2 clearly demonstrate conformational transitions along the Post-Albers cycle of PMCA2. Consequently, they found that the extramembrane domains of PMCA undergo much smaller movements in transition from the E1-Ca/E1-ATP state to the E1-Ca-ATP state and interact with each other with much smaller surface areas in all determined states than those of other P-type ATPases including SERCA1a and SPCA. These structural features seem to nicely explain the uniquely fast transport cycle of PMCA from an energetic perspective. Structures of PMCA in E1 and E2 states suggest that PMCA modulates the affinity for PIP2 depending on the binding/release of Ca²⁺, proposing a role of PIP2 as a latch that promotes the fast Ca²⁺ release and opens a passageway for counterions. Additionally, they found that the PIP2 binding site in PMCA overlaps with the thapsigargin binding site in SERCA, leading to the proposal of a drug-mediated manipulation of intracellular Ca²⁺ level. I appreciate that the authors present new cryo-EM structures of PMCA in eight different states in this single paper and thereby gain structural and mechanistic insights into PIP2-mediated regulation of the PMCA cycle. Meanwhile, I am afraid that some of the authors' conclusions are not fully supported by the present structural data, rendering this paper less convincing. In this regard, the manuscript is still immature. To further reinforce their conclusions and much improve this manuscript, the authors need to address the following major and minor issues appropriately and carefully.

Major issues

1) In most of figures showing structure models, the authors do not display the density map, hence it is hard to judge how precisely they modeled the side chains of amino acid residues involved in Ca²⁺, PIP2, and AMPPNP binding and water molecules located in the PMCA counterion pathway. Indeed, most of cryo-EM structures presented in this work were determined at near-atomic resolutions, which is, in general, not high enough to place the side chains and water molecules accurately. Therefore, I strongly request the authors to display the density map for most of the figure panels. It is also better to additionally show data processing workflows and local resolution maps for all states of PMCA.

2) Related to the above comment, I am concerned that the Ramachandran favored region value is not high enough for most of the intermediate states, suggesting that structure refinement is not perfectly completed for some local regions. Further structure refinement is required except for PMCA-NPTN in E2-Pi state, and figures need to be remade using the further refined models.

3) Structure information on the mode of PIP2 binding in PMCA and its alteration depending on E1 and E2 states is one of the most important findings of this work. However, the lack of density map of the bound PIP2 and its surrounding residues in some figure panels considerably weakens the authors' discussion regarding the PIP2's role as the latch. In this context, it seems better to show possible alterations in the molecular surface structure and electrostatic potential of the PIP2-binding pocket, which may also explain the large difference in affinity for PIP2 between the E1 and E2P states.

4) It would be interesting to additionally discuss the common mechanism of PIP2-mediated regulation of membrane proteins that are already known to be modulated by PIP2, including TRPV and KCNQ1.

5) The authors mentioned that displacement of the D873-Q837 diad results in a large change of the pKa value of D873 (from 3.8 to 7.0). However, it is not described at all how the pKa value of D873 was determined or calculated before and after the displacement. The lack of this information weakens the authors' conclusion of the favored formation of the Ca²⁺-free and protonated states after disruption of the PIP2-N841 interaction.

6) The reason why the authors mutated K344, K347, Q351 to Thr, Thr, Leu, respectively, to reduce the PIP2 binding affinity of PMCA2 is unclear. Looking at the mode of PIP2 binding in PMCA displayed in Fig 5b, these three residues do not seem to tightly interact with PIP2. K344 is not shown in this figure panel. In this connection, the authors' statement "interaction of the PIP2 head group with residue Q351 appeared to be most constant" (line 260-261) is unconvincing. Revision of Fig. 5b and additional explanations about this issue are necessary.

7) Some P-type ATPases have been reported to generate multiple sub-class structures in one intermediate state. In this analogy, I suspect that PMCA-NPTN also adopts different sub-class conformations in some intermediate states. It would be interesting to additionally discuss the possible occurrence of other sub-class conformations and their functional significance in the ultrafast catalytic cycle of PMCA. Conformational overlap at the transition steps may partially explain its fast kinetics.

Minor issues

1) In Fig. 3a, the M8-M9 loop needs to be displayed more clearly to highlight the interaction between PMCA2 and NPTN.

2) In Fig. 3b, two hydrogen bonds that tether Ig2 to Ig3 are unclear.

3) In Fig 4c inset, the side chain of Ser877 is not displayed, hence the involvement of this residue in binding water molecules is unclear.

4) Lines 206-207; This sentence seems imprecise. SERCA actually has sub-micromolar range of Ca²⁺-binding affinity although structures of its Ca²⁺-bound state were solved in the presence of millimolar Ca²⁺.

Referee #3

(Remarks to the Author)

This manuscript examines the transport cycle of the plasma membrane calcium transporter PCMA2. PCMA2s have several

functional distinctions compared the related P2A transporters including their kinetics, their dependence on PIP2 and their requirement for accessory subunits for trafficking to the plasma membrane. Here, the authors present structures of murine PCMA2 by itself or bound to one of accessory subunits, neuropilin-1, in different conformations that together delineate a nearly complete transport cycle. Density corresponding to a PIP2 molecule is resolved in many of these states. Based on these structures, the authors propose models to explain the fast kinetics and the dependency on PIP2. However, the data presented is insufficient to support the proposed models. Combined with the presence of numerous reported structures of P-type ATPases, this study does not represent a sufficient advance to warrant publication in Nature.

Comments.

1) One of the unique features of PCMA2 is its fast transport kinetics. To gain insights in the mechanisms by which PCMA2 is capable of achieving these high transport rates, the authors compare the movements during the transport cycle of PCMA2 with those previously resolved for P2A-ATPases such as SERCA1a, finding that the movements in PCMA2 are smaller. This leads the authors to conclude that energy change associated with the conformational movements in PCMA2 are also smaller. However, such a conclusion cannot be derived from analyses of cryo-EM structures. Cryo-EM structures provide no information on the absolute free energy of a state, nor do they reveal the relative free energy between different states. An alternative experimental approach would be necessary to determine the relative free energies of the various states of PCMA2 in order to compare them with the free energies of the corresponding states of SERCA1a to determine if differences in free energy are sufficient to explain the fast transport kinetics.

2) The authors identify numerous ions, ligands and water molecules in their structures. While the PCMA-NPTN (E2.Pi-AIF4) structure is moderately well resolved (~2.8 Å), all of the other structures are relatively low resolution for resolved non-protein densities (3.0 – 3.6 Å). As the geometric scores are relatively low (Ramachandran allowed > 10%, Bond length RMSD > 0.01 Å) for several of the structures, it is unclear how justified these non-protein densities in the structures. With the exception of the ATP-Mg complex in Supplementary Figure 5, there are no densities presented for any of these molecules in the main text or the supplement. The authors should justify the inclusion of these non-protein molecules in the density map by including figures presenting the raw density around each of the bound non-protein molecules without carving. Additionally, the authors should thoroughly examine their structures to ensure that these geometric issues are supported by the data. Otherwise, the models should be corrected.

1) Continuing from point 2, the authors mention on line 127 that the ATP molecule is better resolved in the PCMA2 E1-ATP structure than in previously a resolved SERCA1a E1-ATP structure and that this indicates that “P2B-ATPases may capture the nucleotide with higher affinity.” Such claims cannot be made from inspection of cryo-EM density maps, particularly with maps at resolutions worse than 3 Å where density maps can have significant artifacts. Direct assessments of binding affinities should be made or this statement should be removed.

3) In all but the E2.Pi structure, the authors observe a bound PIP2 molecule. There are remarkably few interactions with the phosphorylated head group. What provides the specificity of PIP2 over other anionic lipids in the activation of PCMA2? To gain insights into PIP2's role in activation, the authors compare the conformation of the various states in the vicinity of the binding site. In the E1 states, the bound PIP2 interacts with the side chain of N841. In the E2P state, N841 disengages from the PIP2 and instead binds to Q837. The authors propose that this alters the pKa of D873, decreasing the affinity of the Ca²⁺-binding site. However, there is no mention of how the pKa values were calculated for D873 in the methods. Nor are there any direct tests of the effect of Q837 on the binding affinity of Ca in the various states. The authors should describe how the pKa calculations were performed and evaluate how precise are the calculations at the intermediate resolutions of the structures presented. Was the pKa of any other residue significantly altered? The authors should also perform binding studies in different states for wild-type and transporters in which Q837 and D873 are mutated to directly test their hypothesis for the role of PIP2 in transport.

4) How was thapsigargin docked into the structures and how was the pose validated? A more detailed analysis and description should be included in the methods.

5) The figure legends are vague in several places and need to be clarified to aid the reader. For example, it is not stated to which state the white and colored by domain structures in Figure 2d correspond. Please clarify the legends to aid the reader.

Minor comments.

1) The structures are presented in the opposite orientation from what would be expected for a transmembrane protein with the extracellular side facing down. Is there a particular reason why they are presented in this manner?

2) The movements depicted in Figure 2 are not clear. To what do the different colors correspond to in panels c-f? It is not listed in the Figure Legends. For panel d, it would be helpful to color the white state as a low intensity version of the colored state to help the reader compare which domains correspond to which. Similarly, the labels in panels e and f are inconsistently placed making it hard to compare the two states.

3) What are the “other inter-residue contacts” in Figure 3a? Do these stabilize the interaction between PCMA2 and NPTN? There are so many that it is difficult to visualize in the figure. If they contribute to stabilizing the interface, then they should be better described and presented.

4) The resolutions of the presented structures are too low for inter-atom distances to be presented with accuracy to the hundredth of an Å in Figure 5.

Version 1:

Reviewer comments:

Referee #2

(Remarks to the Author)

After careful reading of the revised manuscript and rebuttal letter, I found that while the authors responded to some of my critical comments appropriately, the paper seems to require further revisions before recommendation for publication in this journal as described below.

1) Regarding Reply 2.1, I do not agree at all with the authors' statement that "in the case of complex strictures such as PMCA, it is difficult to show densities and models superimposed plus side chain labels without losing clarity". There should be some ways to clearly highlight local structures with density maps and models superimposed even when molecular size of a target protein is quite huge, as actually done by the authors in supplemental Fig. 10. In this regard, revised Fig. 3c & d, Fig. 4c and Fig. 5d are still unacceptable. In particular, the positions and orientations of the Q351, N841, Q837 and D873 side chains seem critical to discuss the mode of interaction between PIP2 and PMCA2 and the PIP2-mediated activation of Ca²⁺ transport by PMCA2. Similarly, to discuss the counterproton binding sites and water-lined release pathway in PMCA2, the density needs to be shown clearly for the side chains of Q835, S877 and R969 as well as water molecules in Fig. 4c. Owing to the lack of well-resolved density maps for the residues above, the authors did not address my comments 2.1, 2.8, 2.9, and 2.10 in a well convincing manner.

2) Regarding Reply 2.5, it is still hard to understand why the pKa value of D873 largely changes from 3.8 to 7.0, which seems inconsistent with the values displayed in supplementary Table 6 (6.07 in E1 to 6.9 in E2P).

3) Page 8, line 207: The absence of site I in SPCA1 is first reported by Chen et al. *Sci. Adv.* (2023) eadd9742. This paper should be cited.

Referee #3

(Remarks to the Author)

Overall, the manuscript is greatly improved in its revised form. Many of the issues with the structures and their presentations have been corrected. If a few minor issues (listed below) can be corrected, the manuscript would be suitable for publication.

Figure 2c remains hard to interpret. The four domains move in distinct ways and it is not clear from the figure which domain is moving in which way. Perhaps a vector diagram depicting the per-residue or per secondary structural element movements may aid the reader in visualizing this ensemble of complex movements. Also, the transmembrane domain of the E1-Ca-ATP state is grey in panel d. It would be helpful to keep the coloring consistent.

In Figure 3b, the dashed cyan line is described as a hydrogen bond, but it is not clear which residue, along with E348, participates in the interaction. In Figure 2c, it is not clear if the side chain or backbone of S936 interacts with R244. These details should be clarified.

In Supplementary Figure 14, the PIP2 molecule is oriented oppositely from in the panels c-e. Please make the orientation consistent.

Version 2:

Reviewer comments:

Referee #2

(Remarks to the Author)

The authors have appropriately addressed all of my comments. I can now recommend this version of the manuscript for publication in this journal.

Referee #3

(Remarks to the Author)

The revised version has addressed my concerns and is now suitable for publication.

Point-to-point response to the reviewers' comments

We thank all reviewers for the overall positive reception of the two manuscripts. Guided and inspired by the constructive criticism and direct suggestions, we made a concerted effort to strengthen our conclusions through the complete reworking of the text and figures.

Structural paper (Vinayagam et al.):

Reviewer #1

My comments regarding the manuscript “Molecular Mechanism...” by Vinayagam et al are confined solely to the use of BK channels expressed in COS cells to monitor the ability of WT or manipulated plasma membrane Ca²⁺-ATPase to reduce cytosolic Ca²⁺. Because BK channels are regulated by both Ca²⁺ and voltage, in whole cell recordings determination of the membrane potential at which BK channels are half activated provides a fairly accurate estimate of the actual cytosolic Ca²⁺ concentration. Based on results presented in the companion paper, the authors demonstrated that coexpression of PMCA-NPTN effectively and dramatically can reduce a nominal cytosolic Ca²⁺ concentration of 10 micromolar, down to near 100 nM. With this assay, the authors can then potentially manipulate PMCA2 function allowing results that may inform their structures.

Fig 3d clearly demonstrates that whether PMCA2 is expressed alone or with NPTN, the clearance of Ca²⁺ from the COS cells is similar.

Fig. 4 provides structures of two disease-associated mutant PMCA2 transporters and Fig. 4b nicely confirms that both mutants that are evaluated (E412K and S877F) are hindered in their ability to remove Ca²⁺, with E412K interfering with Ca²⁺ coordination and S877F altering counterion transport.

Figure 5 examines the requirement for PIP₂ in PMCA2 function, and the authors use two functional experiments to validate the impact of manipulations of PIP₂ on their BK-COS Ca²⁺ clearance assay. In Fig. 5a, using the Ciona voltage-sensitive phosphatase which rapidly reduces membrane concentrations of PIP₂, the authors show that with CiVSP stimulation produces a profound loss of ability to PMCA2 to clear Ca²⁺.

[1.1] My only concern here is that in the text the authors mention “this was done using co-expressed PIP₂-phosphatase CiVSP that effectively depletes PIP₂ after activation of its enzymatic activity (Methods).” Unfortunately, there appears to be no mention in the Methods or in the figure legends of how CiVSP was activated.

Reply 1.1: CiVSP is a voltage-dependent PIP₂-phosphatase that is activated by membrane depolarization. We have added this statement to the revised manuscript (text, legend to Fig. 5 and Methods).

In Fig. 5B, the authors examine functional impact of two mutations which alter PIP2 interactions in their structures. Both of these mutations that disrupt PIP2 binding result in reduced ability of the mutant PCMA2 proteins to clear Ca²⁺.

Finally, the authors explore the possibility that thapsigargin, an inhibitor of SERCA mediated Ca²⁺- uptake, may also interact with PMCA2. This is supported by docking computational tests, but apparently not structural data. The hypothesis that thapsigargin does interact with PMCA2 is supported by functional data in Fig. 6c that clearly shows that application of thapsigargin to cells expressing PMCA2 hinders the clearance of Ca²⁺ in a nicely concentration dependent fashion.

Overall, the functional tests are straight-forward and seem to nicely complement the structures.

We thank this reviewer for this nice remark.

Referee #2

In this work, the authors report cryo-EM structures of plasma membrane Ca²⁺-ATPase (PMCA) complexed with its functional partner NPTN in 6 different states and of PMCA alone in two different states. In all cryo-EM density maps but that of the E2Pi state, a density corresponding to PIP2, a modulator of PMCA, was observed. They also conducted functional assays based on the structural information by monitoring the PMCA-mediated Ca²⁺ extrusion using Ca²⁺-activated BK channels as reporters. The totally eight cryo-EM structures of PMCA2 clearly demonstrate conformational transitions along the Post-Albers cycle of PMCA2. Consequently, they found that the extramembrane domains of PMCA undergo much smaller movements in transition from the E1-Ca/E1-ATP state to the E1-Ca-ATP state and interact with each other with much smaller surface areas in all determined states than those of other P-type ATPases including SERCA1a and SPCA. These structural features seem to nicely explain the uniquely fast transport cycle of PMCA from an energetic perspective. Structures of PMCA in E1 and E2 states suggest that PMCA modulates the affinity for PIP2 depending on the binding/release of Ca²⁺, proposing a role of PIP2 as a latch that promotes the fast Ca²⁺ release and opens a passageway for counterions. Additionally, they found that the PIP2 binding site in PMCA overlaps with the thapsigargin binding site in SERCA, leading to the proposal of a drug-mediated manipulation of intracellular Ca²⁺ level. **I appreciate that the authors present new cryo-EM structures of PMCA in eight different states in this single paper and thereby gain structural and mechanistic insights into PIP2-mediated regulation of the PMCA cycle.**

We thank this reviewer for their appreciation of our work.

Meanwhile, I am afraid that some of the authors' conclusions are not fully supported by the present structural data, rendering this paper less convincing. In this regard, the manuscript is still immature. To further reinforce their conclusions and much improve this manuscript, the authors need to address the following major and minor issues appropriately and carefully.

We apologize that the previous presentation of our work was not sufficient to convince this reviewer of the importance of our study and thank for the critical remarks. We thoroughly revised the manuscript and figures, addressed all major and minor issues and we are sure that this reviewer will have no more technical concerns about our data and will be convinced by the revised manuscript.

Major issues:

[2.1] In most of figures showing structure models, the authors do not display the density map, hence it is hard to judge how precisely they modeled the side chains of amino acid residues involved in Ca²⁺, PIP₂, and AMPPNP binding and water molecules located in the PMCA counterion pathway. Indeed, most of cryo-EM structures presented in this work were determined at near-atomic resolutions, which is, in general, not high enough to place the side chains and water molecules accurately. Therefore, I strongly request the authors to display the density map for most of the figure panels. It is also better to additionally show data processing workflows and local resolution maps for all states of PMCA.

Reply 2.1: In the case of complex structures such as PMCA, it is usually difficult to show densities and models superimposed plus side chain labels without losing clarity. Therefore, we had mainly shown the models in the figures and offered to provide the densities and models for detailed inspection by the reviewers. However, this obviously did not happen and we apologize for this. However, we have used the criticism constructively and are now showing relevant densities in the figures (Figure 1a, 2a, 3a, 4b,c, Supplementary Figure 1-5, 8, 10, 15). We have also added short movies (Supplementary Videos 2-5) for better visualization. In addition, we provide now data processing workflows and local resolution maps for all states of PMCA in the revised manuscript (Supplementary Figure 1-5).

It is important to emphasize that we state overall resolutions. The local resolutions in the transmembrane regions, however, are markedly higher and enabled the accurate modeling of side chains, the phospholipid PIP₂ and ordered water molecules. The local resolutions are now shown in Supplementary Figures 1-5.

In general, we have completely reworked our figures to make it easier for the reader to understand our described mechanisms.

[2.2] Related to the above comment, I am concerned that the Ramachandran favored region value is not high enough for most of the intermediate states, suggesting that structure refinement is not perfectly completed for some local regions. Further structure refinement is required except for PMCA-NPTN in E2-Pi state, and figures need to be remade using the further refined models.

Reply 2.2: This reviewer is right and the values were indeed not satisfactory. We went back and re-ran the refinements and remade all figures. Now the models are improved with Ramachandran favored region values of 94-96 % as is reported in Supplementary Table 2.

[2.3] Structure information on the mode of PIP₂ binding in PMCA and its alteration depending on E1 and E2 states is one of the most important findings of this work. However, the lack of density map of the bound PIP₂ and its surrounding residues in some figure panels considerably weakens the authors' discussion regarding the PIP₂'s role as the latch. In this context, it seems better to show possible alterations in the molecular surface structure and electrostatic potential of the PIP₂-binding pocket, which may also explain the large difference in affinity for PIP₂ between the E1 and E2P states.

Reply 2.3: We thank this reviewer for this comment. We have reworked Figure 5, Supplementary Fig. 14 and Supplementary Fig. 15. They show now the densities for PIP₂ at the lipid-binding site in all PMCA2-NPTN models, as well as the electrostatic potential of the PIP₂-binding pocket as suggested by this reviewer. The figures reveal that the structure of the PIP₂ binding pocket changes in particular in the E2.P_i state where it is more open. Since the charge of the binding pocket does not change significantly, electrostatics cannot be the only determinant for the specificity of PIP₂-binding and shape complementarity appears to be another important factor for the specific binding of PIP₂ to PMCA2. See also **Reply 2.6**. We describe this now in the revised manuscript.

[2.4] It would be interesting to additionally discuss the common mechanism of PIP₂-mediated regulation of membrane proteins that are already known to be modulated by PIP₂, including TRPV and KCNQ1.

Reply 2.4: This is a nice suggestion. We have added a respective chapter to the Discussion section of the revised manuscript.

[2.5] The authors mentioned that displacement of the D873-Q837 diad results in a large change of the pK_a value of D873 (from 3.8 to 7.0). However, it is not described at all how the pK_a value of D873 was determined or calculated before and after the displacement. The lack of this information weakens the authors' conclusion of the favored formation of the Ca²⁺-free and protonated states after disruption of the PIP₂-N841 interaction.

Reply 2.5: We have calculated the pK_a values using the program PROPKA. We describe this now in the Data analysis subsection in Methods and cite the relevant paper.

[2.6] The reason why the authors mutated K344, K347, Q351 to Thr, Thr, Leu, respectively, to reduce the PIP₂ binding affinity of PMCA2 is unclear. Looking at the mode of PIP₂ binding in PMCA displayed in Fig 5b, these three residues do not seem to tightly interact with PIP₂. K344

is not shown in this figure panel. In this connection, the authors' statement "interaction of the PIP₂ head group with residue Q351 appeared to be most constant" (line 260-261) is unconvincing. Revision of Fig. 5b and additional explanations about this issue are necessary.

Reply 2.6: We revised Figure 5 as suggested by this reviewer and also added surfaces depicting the electrostatic potential (see **Reply 2.3**). It becomes clear from the reworked Figure 5b and Supplementary Fig. 14f that while Q351 interacts with a phosphate group of PIP₂, K344 and K347 are crucial for creating the positively charged vestibule of the PIP₂ binding pocket although they do not tightly interact with PIP₂. Site-directed mutagenesis combined with functional recordings confirmed the significance of these three residues (Figure 5e).

In general, we propose that four features of the PIP₂-binding site together render it specific for this particular lipid: 1. Shape complementarity – PIP₂ fits ideally into the binding pocket in the PIP₂-bound states. In the states without PIP₂ it is much wider. 2. Positively charged vestibule – which is important for stably binding the negatively charged head groups of the lipid. 3. Direct interaction of Q351 with a phosphate group of PIP₂. 4. Hydrophobicity – the hydrophobic patch in the binding pocket is important for the stable binding of the lipid tail.

This is described in detail in the revised manuscript.

[2.7] Some P-type ATPases have been reported to generate multiple sub-class structures in one intermediate state. In this analogy, I suspect that PMCA-NPTN also adopts different sub-class conformations in some intermediate states. It would be interesting to additionally discuss the possible occurrence of other sub-class conformations and their functional significance in the ultrafast catalytic cycle of PMCA. Conformational overlap at the transition steps may partially explain its fast kinetics.

Reply 2.7: We thank the reviewer for pointing this out. We indeed observe heterogeneity in our reconstructions of PMCA2. In particular, density for the cytoplasmic A-domain is weak in some of the reconstructions, indicating flexibility. However, additional 3D classifications did not reveal any discrete sub-class conformations. This indicates that the A-domain may undergo continuous motion, which is difficult to analyze. Importantly, we do not observe structural heterogeneity in the transmembrane region (M-domain) within the PMCA2 reconstructions.

Minor issues

[2.8] In Fig. 3a, the M8-M9 loop needs to be displayed more clearly to highlight the interaction between PMCA2 and NPTN.

Reply 2.8: We agree with this observation. We have updated the figure and now show a zoom of this interaction in Figures 3b, c.

[2.9] In Fig. 3b, two hydrogen bonds that tether Ig2 to Ig3 are unclear.

Reply 2.9: We thank this reviewer for pointing this out. In the updated Figure 3, a zoom of the interaction between Ig2 and Ig3 is shown in subpanel (d). We believe that the extra panel now clearly displays the interaction.

[2.10] In Fig 4c inset, the side chain of Ser877 is not displayed, hence the involvement of this residue in binding water molecules is unclear.

Reply 2.10: We have updated the figure, now displaying the side chain of S877. The water molecules do not bind directly to S877, but are in its immediate vicinity. Therefore, a mutation of S877 probably changes the entire electrostatics of this region and its ability to bind water molecules and conduct protons. We describe this now more clearly in the revised manuscript.

[2.11] Lines 206-207; This sentence seems imprecise. SERCA actually has sub-micromolar range of Ca²⁺-binding affinity although structures of its Ca²⁺-bound state were solved in the presence of millimolar Ca²⁺.

Reply 2.11: This is correct and we have changed the sentence accordingly.

Reviewer #3

This manuscript examines the transport cycle of the plasma membrane calcium transporter PCMA2. PCMA2s have several functional distinctions compared the related P2A transporters including their kinetics, their dependence on PIP2 and their requirement for accessory subunits for trafficking to the plasma membrane. Here, **the authors present structures of murine PCMA2 by itself of bound to one of accessory subunits, neuroplastin, in different conformations that together delineate a nearly complete transport cycle. Density correspond to a PIP2 molecule is resolved in many of these states. Based on these structures, the authors propose models to explain the fast kinetics and the dependency on PIP2.**

We thank this reviewer for appreciating the scope and complexity of our work.

However, the data presented data is insufficient to support the proposed models. Combined with the presence of numerous reported structures of P-type ATPases, this study does not represent a sufficient advance to warrant publication in Nature.

We apologize that the previous presentation of our work was not sufficient to convince this reviewer of the importance of our study and thank for the critical remarks. We thoroughly revised the manuscript and figures and we are sure that this reviewer will have no more technical concerns about our data and will be convinced by the revised manuscript.

Regarding the importance of our structures: We present the first complete transport cycle of a plasma membrane Ca^{2+} -ATPase and elucidate the structural underpinnings for both its ultra-fast transport kinetics and its obligatory hallmark regulation by the phospholipid PIP_2 . Moreover, the work introduces the PIP_2 -binding site as an unrecognized drug-able site for control of plasma membrane Ca^{2+} transport. Importantly, this would not have been possible by the already determined structures of other P-type ATPases. Together with the complementary manuscript Constantin et al., our study is of fundamental importance for the broad field of cell biology, neurobiology and medicine and, in our opinion, warrants publication in *Nature*.

Comments:

[3.1] One of the unique features of PCMA2 is its fast transport kinetics. To gain insights in the mechanisms by which PCMA2 is capable of achieving these high transport rates, the authors compare the movements during the transport cycle of PCMA2 with those previously resolved for P2A-ATPases such as SERCA1a, finding that the movements in PCMA2 are smaller. This leads the authors to conclude that energy change associated with the conformational movements in PCMA2 are also smaller. However, such a conclusion cannot be derived from analyses of cryo-EM structures. Cryo-EM structures provide no information on the absolute free energy of a state, not do they reveal the relative free energy between different states. An alternative experimental approach would be necessary to determine the relative free energies of the various states of PCMA2 in order to compare them with the free energies of the corresponding states of SERCA1a to determine if differences in free energy are sufficient to explain the fast transport kinetics.

Reply 3.1: There is an obvious misconception, since we did not make any conclusions on differences in free energy states (which would anyway not be relevant for estimating or comparing kinetics!). Rather, transition kinetics between states are determined by their transitional energy barriers that largely depend on the number and free energies of residue interactions to be broken and bond conformations to be altered to allow for the respective transitions. We, therefore, investigated both, the interaction surface between the cytoplasmic domains as well as the extent of their movement and found that both were significantly smaller in PMCA than in SERCA suggesting lower transitional energy barriers which would of course be essential for a higher transport rate.

[3.2] The authors identify numerous ions, ligands and water molecules in their structures. While the PCMA-NPTN (E2.Pi-AlF4) structure is moderately well resolved ($\sim 2.8 \text{ \AA}$), all of the other structures are relatively low resolution for resolved non-protein densities ($3.0 - 3.6 \text{ \AA}$). As the geometric scores are relatively low (Ramachandran allowed $> 10\%$, Bond length RMSD $> 0.01 \text{ \AA}$) for several of the structures, it is unclear how justified these non-protein densities in the structures. With the exception of the ATP-Mg complex in Supplementary Figure 5, there are no densities presented for any of these molecules in the main text or the supplement. The authors should justify the inclusion of these non-protein molecules in the density map by including figures presenting the raw density around each of the bound

non-protein molecules without carving. Additionally, the authors should thoroughly examine their structures to ensure that these geometric issues are supported by the data. Otherwise, the models should be corrected.

Reply 3.2: In the case of complex structures such as PMCA, it is usually difficult to show densities and models superimposed plus side chain labels without losing clarity. Therefore, we had mainly shown the models in the figures and offered to provide the densities and models for detailed inspection by the reviewers. However, this obviously did not happen and we apologize for this. However, we have used the criticism constructively and are now showing relevant densities in the figures (Figure 1a, 2a, 3a, 4b,c, Supplementary Figure 1-5, 8, 10, 15). We have also added short movies (Supplementary Videos 2-5) for better visualization. In addition, we provide now data processing workflows and local resolution maps for all states of PMCA in the revised manuscript (Supplementary Figure 1-5).

It is important to emphasize that we state overall resolutions. The local resolutions in the transmembrane regions, however, are markedly higher and enabled the accurate modeling of side chains, the phospholipid PIP₂ and ordered water molecules. The local resolutions are now shown in Supplementary Figures 1-5.

In general, we have completely reworked our figures to make it easier for the reader to understand our described mechanisms.

This reviewer is right and the refinement values were indeed not satisfactory. We went back and re-ran the refinements and remade all figures. Now the models are improved with Ramachandran favored region values of 94-96 % as is reported in Supplementary Table 2.

[3.3] Continuing from point 2, the authors mention on line 127 that the ATP molecule is better resolved in the PCMA2 E1-ATP structure than in previously a resolved SERCA1a E1-ATP structure and that this indicates that “P2B-ATPases may capture the nucleotide with higher affinity.” Such claims cannot be made from inspection of cryo-EM density maps, particularly with maps at resolutions worse than 3 Å where density maps can have significant artifacts. Direct assessments of binding affinities should be made or this statement should be removed.

Reply 3.3: We agree with the reviewer that structural data alone cannot be used to draw strong conclusions regarding quantitative differences in nucleotide-binding affinities between PMCA2 and SERCA1a.

Nevertheless, in our cryo-EM structure of PMCA2 in the E1-ATP state, we observe strong, unambiguous density for the bound nucleotide in the absence of Ca²⁺. In contrast, in crystal structures of SERCA1a in the E1-ATP state, density for the added nucleotide is extremely weak (Supplementary Figure 8). This suggests that PMCAs can stably associate with the nucleotide already before Ca²⁺ binds, while SERCAs cannot, which is another indication for the fast transport rates of PMCA. We rephrased this part of the manuscript accordingly.

[3.4] In all but the E2.Pi structure, the authors observe a bound PIP₂ molecule. There are remarkably few interactions with the phosphorylated head group. What provides the specificity of PIP₂ over other anionic lipids in the activation of PCMA2? To gain insights into PIP₂'s role in activation, the authors compare the conformation of the various states in the vicinity of the binding site. In the E1 states, the bound PIP₂ interacts with the side chain of N841. In the E2P state, N841 disengages from the PIP₂ and instead binds to Q837. The authors propose that this alters the pK_a of D873, decreasing the affinity of the Ca²⁺-binding site. However, there is no mention of how the pK_a values were calculated for D873 in the methods. Nor are there any direct tests of the effect of Q837 on the binding affinity of Ca in the various states. The authors should describe how the pK_a calculations were performed and evaluate how precise are the calculations at the intermediate resolutions of the structures presented. Was the pK_a of any other residue significantly altered? The authors should also perform binding studies in different states for wild-type and transporters in which Q837 and D873 are mutated to directly test their hypothesis for the role of PIP₂ in transport.

Reply 3.4: In general, we propose that four features of the PIP₂-binding site together render it specific for this particular lipid: 1. Shape complementarity – PIP₂ fits ideally into the binding pocket in the PIP₂-bound states. In the states without PIP₂ it is much wider. 2. Positively charged vestibule – which is important for stably binding the negatively charged head groups of the lipid. 3. Direct interaction of Q351 with a phosphate group of PIP₂. 4. Hydrophobicity – the hydrophobic patch in the binding pocket is important for the stable binding of the lipid tail. This is described in detail in the revised manuscript.

We have calculated the pK_a values using the program PROPKA. D873 and E412 are the only ionizable residues near the PMCA2 ion binding site that could be selected for pK_a analysis. We describe this now in the Data analysis subsection in Methods and cite the relevant paper. All residues described here are well resolved in our structures (Supplementary Fig. 15a). We are therefore confident that our calculations are based on a good empirical foundation.

We probed the obligatory role of PIP₂ in Ca²⁺ transport by PMCA-NPTN directly by removal of the phospholipid under 'in-situ'-conditions via the PIP₂-specific phosphatase CiVSP (before and after activation of the enzyme): Removal of PIP₂ entirely abolished the fast transport by the Ca²⁺-ATPase as shown here for the first time (Figure 5a). We verified the phospholipid-action by structure-guided mutagenesis effectively attacking the two major sites of interaction. Separate alteration of both sites, (i) N841Q (with the ester-bond of PIP₂) and (ii) the positively charged vestibule of the binding pocket formed by the amino-groups of Q351, K347 and K344 (with phosphate groups of PIP₂), largely decreased transport activity (Figure 5e). Binding studies are experimentally impossible.

[3.5] How was thapsigargin docked into the structures and how was the pose validated? A more detailed analysis and description should be included in the methods.

Reply 3.5: We have added a more detailed analysis and description in the methods.

[3.6] The figure legends are vague in several places and need to be clarified to aid the reader. For example, it is not stated to which state the white and colored by domain structures in Figure 2d correspond. Please clarify the legends to aid the reader.

Reply 3.6: We have revised all figure legends. We believe that they now clearly describe what is shown in the figures.

Minor comments:

[3.7] The structures are presented in the opposite orientation from what would be expected for a transmembrane protein with the extracellular side facing down. Is there a particular reason why they are presented in this manner?

Reply 3.7: The reviewer is correct that most membrane protein structures are presented with the extracellular side facing up. However, in the P-type ATPase field, it is more common to display the reverse arrangement with the extracellular side facing down (see e.g., PMID: 23455424, 37258749). We have prepared our figures according to these standard practices in the field.

[3.8] The movements depicted in Figure 2 are not clear. To what do the different colors correspond to in panels c-f? It is not listed in the Figure Legends. For panel d, it would be helpful to color to white state as a low intensity version of colored state to help the reader compare which domains correspond to which. Similarly, the labels in panels e and f are inconsistently placed making it hard to compare the two states.

Reply 3.8: We have reworked Figure 2. We clearly explain the colors now in the legend and checked the placement of labels in the panels. Instead of displaying an overlay of the structures in panel d (now c) which was indeed difficult to recognize, we show them now next to each other in two panels. In addition, we have added Supplementary Videos 2 and 3 where we show morphs between the structures to better visualize the conformational differences between states.

[3.9] What are the “other inter-residue contacts” in Figure 3a? Do these stabilize the interaction between PCMA2 and NPTN? There are so many that it is difficult to visualize in the figure. If they contribute to stabilizing the interface, then they should be better described and presented.

Reply 3.9: This statement was indeed confusing and has been removed. In the updated Figure 3, we provide several zooms of the interaction interface, which makes it easier to understand the interaction between PMCA2 and NPTN.

[3.10] The resolutions of the presented structures are too low for inter-atom distances to be presented with accuracy to the hundredth of an Å in Figure 5.

Reply 3.10: We agree and have adjusted it accordingly.

Functional paper (Constantin et al.):

Reviewer #1

~~This paper provides important progress towards understanding the role of one Ca^{2+} extrusion process, that mediated by PMCA2 in association with neuroplastin (NPTN)..... Rates of Ca transport of that magnitude are a bit of a surprise for PCNMA2, although the authors nicely address why this may not have been noted before. That PCNMA2 clears Ca with such rapidity is an important advance in regards to understanding the molecular underpinnings of cytosolic Ca^{2+} clearance.~~

~~Most of my concerns below pertain to omissions in experimental details, details of analysis, and/or suggestions that may help strengthen some aspects of the paper.~~

~~**[1.1]** It might be useful for readers to know that PMCA is not thought to be electrogenic. In that vein, what extent of acidification might be expected given the high Ca transport rates and likely exchange of 2 protons for each Ca? Might the rightward BK gating shifts that are produced by acidification contribute to the appearance of rapid decay of BK current?~~

~~**Reply 1.1:** Intracellular acidification resulting from the Ca^{2+} 2H^{+} counter-transport by PMCA is counteracted by 5 mM HEPES in the pipette solution effectively buffering the pH in the physiological range. Acidification of the intracellular milieu (in contrast to the extracellular) leads to a left-shift in BK_{Ca} activation (eg. Biophysical Journal, 84, 2969), an increased deactivation speed may, therefore, not be expected.~~

~~**[1.2]** In Fig. 1b, if ATP is omitted from the pipette solution in cells expressing PCMA2-NPTN, does that abolish the apparent ability of PCMA2 to reduce cytosolic Ca?~~

~~**Reply 1.2:** The respective ATP washout experiment (entirely removing transport activity of PMCA-NPTN) was presented in our previous paper in Neuron (96, 827-838; Fig. 6B).~~

~~**[1.3]** In Fig. 1b, the ability of PMCA to remove the 10 μM Ca presumably depends to some extent on the flux rate of CaEGTA and free Ca from the pipette into the cell. Given knowledge of the pipette diameter, the flux of total Ca^{2+} into the cell per second can presumably be calculated. Since you know the numbers of PCMA2-NPTN per cell, it might be interesting to see whether the transport rates and # of PMCA2 molecules match with the entry from the pipette.~~

~~**Reply 1.3:** The reviewer's assumptions are correct and the calculated Ca^{2+} entry from the pipette exactly matches the Ca^{2+} outflux through the PMCA2 molecules.~~

~~**[1.4]** Figures 1b-d are very compelling results!!!! Congrats!~~

~~**Reply 1.4:** Comment is well appreciated.~~

~~[1.5] There are details in the GCaMP6 records that are curious. The reason given by the authors for the slower decay makes sense, but it would have been nice to see the rising phase with the same time base and stimulation protocol as in 1e (Perhaps it was, but this wasn't clear in the figure). Is there a reason why there isn't some slow rise in the GCaMP6 signal? It would be good to see the 1 s calibration label given on the panel.~~

~~Reply 1.5: The experiments with the GCaMP6 dye (Fig. 1e) used identical conditions as the experiments in Fig. 1c. The minor differences in time course (in the first 50 ms following Ca^{2+} influx) are due to the gating kinetics of the BK_{Ca} channels. Labelling of the scale bar was moved from the legend to the figure as suggested by the reviewer.~~

~~[1.6] Figure 2. In the bar plots, isn't it standard practice now to also show individual values that go into each mean? This reader is not knowledgeable about issues that may confound this kind of particle counting, but presumably there could be particles that are incorrectly identified (non-specificity of Abs) and others that are not labelled. Are there ways to give readers a sense of how quantitative this method is? Might a useful additional control be doing the same analysis on BK channels? In that case, one has the benefit that total membrane density of functional BK channels on the cell surface can independently be made from estimates of maximal BK conductance~~

~~Reply 1.6: Target staining in freeze fracture replicas combined with (gold)particle counting (in EM pictograms) is the most accurate technology available for quantification of membrane proteins. BK_{Ca} channels were used as reporters whose density is not meaningful for the purpose of this study.~~

~~The antibodies used for labelling of PMCA and Cav2.2 were tested for target specificity and background using target knockout tissue.~~

~~Individual data points were added in Figures 2 and 4 as advised by the reviewer.~~

~~[1.7] The simulations in figure 3 that are used to correlate potential microscopic Ca extrusion rates to data are important, but several aspects of this analysis seem glossed over. It would be important that the authors more explicitly specify the following:~~

~~a. Which model is used in 3a and which model in 3b.~~

~~Reply (a): Steady state activation curves used the stationary model (Fig. 3a, parameters detailed in Suppl. Tables 1 and 2), pulse experiments use the dynamical model) in Fig. 3b, Suppl. Table 3). This was made more explicitly in the revised ms.~~

~~b. What voltage was used for the simulations in 3b.~~

~~Reply (b): Computation modelling used the identical voltage protocol as the experiment in Fig. 1e~~

e. Although activation curves in 3a indicate a maximal G_{max} for the steady state case, the “rel. BK current” in 3b provides no measure of the fractional activation of BK current for the case that is being considered. This concern also applies to the evaluation of the model in Extended Data Figure 5. Although the correspondence in time course seems clear, unless both simulated and experimentally measured traces are normalized against the maximal activatable current, we don't know whether both traces represent the same extent of fractional activation. The vertical calibration bars could ideally be given as fractional activation at the given voltage.

Reply (e): As assumed by the reviewer, the fractional current(s) in Fig. 3b and ED Fig.5 were ‘normalized’ to the maximal current(s) observed in the experiments.

d. Although the authors do not comment on it, the effective /s PMCA transport rates that affect the shift in GV_s as a function of Ca (3a) vs. the effect on clearance rate (3b) cover 4-5 fold different ranges. Whereas in 3b, 20,000/s approximates the observed effect of PMCA expression, in 3a less than 5000/s appears able to approximate the PMCA situation. There may be reasons why one would expect such differences in comparing a steady state to a rate-based measure, but this is not addressed. One wonders whether the apparent use of different models with different parameters might contribute to these differences. Having said this, irrespective of the differences, whether the precise microscopic rate is 4000/s or 20000/s, the results and analysis support the idea that these rates greatly exceed those previously reported for PMCA.

Reply (d): The computational modelling in Fig. 3a provides a lower limit for the transport rate (of at least 5 kHz !), as the sensitivity of the steady state activation curves (Fig. 1b and experimental trace in Fig. 3a) is limited. In contrast, the pulse experiments (Fig. 1c) allow for a more precise estimate of the pump rate, yielding values of about 20 kHz. Accordingly, the difference is not due to different models, but rather to different resolutions of the different experimental setups.

[1.8] There are aspects of the models presented in SI2 and SI3 that need elaboration. The parameters listed in SI2 and SI3, although purported to be modifications of Cox, Cui, & Aldrich, 1997, appear to be quite different from those in Cox et al., although the way the parameters are expressed confuses the issue. It may not matter exactly what the rates are, as long as unbinding of Ca²⁺ from BK channels is rapid enough to report the changes in cytosolic Ca. However, those with extensive knowledge about BK will find the nanomolar binding affinities for the 2nd and 4th Ca binding step to seem rather odd. Most BK knowledgeable readers will recognize that any 10 state model of BK is only a rough approximation, but that even so it can still be a useful descriptor of fractional activation as a function of stimulation conditions. However, to call Cox, Cui and Horrigan an established model seems a bit of an overstatement, since that model has been revised and amended many times over the years. I don't think that matters in terms of the present paper. All that seems necessary here is that you have a reasonable approximation that, in particular, captures the deactivation of BK current when Ca is decreased or the voltage is made more negative.

Reply 1.8: We fully agree with the reviewer here.

~~Also related to this topic, it is never made clear why there is a need for both the steady-state model and the dynamic model, since presumably the dynamic model when Ca is fixed at either 0.1 or 10 μM will yield steady-state predictions. Why shouldn't these correspond? However, each model has very different net voltage dependencies, which would result in very different predicted steady-state GV curves for each. One wonders why that was necessary and if terms are "adjusted" simply to make simulations match data contingent upon the particular measurement being made.~~

~~**Reply 1.8:** As stated by the reviewer, steady state experiments can be equivalently described by the dynamical model (converging on the stationary solution after a sufficiently long time with Ca^{2+} and voltage fixed). The reason for using both models is the different mathematical solution: For stationary experiments we solved the stationary diffusion equation with finite element methods, while for the dynamic pulse experiments we numerically integrated the time-dependent diffusion equation. Moreover, we decided to fit the dissociation constants for the stationary model independently from the rate constants for the dynamic model because two completely different data from different experimental setups were used for calibration. Nevertheless, both models predict nearly identical steady-state GV curves.~~

~~There seem to be two problems with parameter values. In SI2 for the steady-state simulation, the model lists the $C_0 \rightleftharpoons C_1$ constant as $K_{C1}[e]$. Since K_{C1} is $27.1 \mu\text{M}$, the calculated constant will then be in units of $(\mu\text{M})^2$, which makes no sense. Perhaps the C_1/C_0 ratio is $[C]/K_{C1}$.~~

~~**Reply 1.8:** The dissociation constants $K_{Ci}[e]$ and $K_{Oi}[e]$ shown were meant to reflect dependence of state transitions (between C_1 and O_1 states) on the Ca^{2+} concentration (not multiplication); to avoid misinterpretation we changed the constants to K_{Ci} and K_{Oi} in the revised Suppl. Table 2.~~

~~In the dynamic model, Q_1 of 0.225 is assigned to the voltage-sensing deactivation rate, while $Q_2 = -0.11$ is assigned to activation. It seems that either the signs or the values are likely reversed.~~

~~**Reply 1.8:** Please note that for the voltage-dependent transition rates (and the charges) the definition $e_i(V) = \exp(+Q_i FV/RT)$ is used in Suppl. Table 3 (while Cox, Cui, & Aldrich, 1997 used $\exp(-Q_i FV/RT)$).~~

~~**[1.9]** In the discussion of the previous estimates of low PMCA transport rates, would it be useful to mention such information as the typical PMCA affinity for Ca^{2+} (100–200 nM range), and that interaction with CaM may increase affinity and transport rates? Presumably, the COS cells have plenty of available CaM.~~

~~**Reply 1.9:** CaM concentrations in all epithelial cells is in the mM range; nonetheless, we have never observed CaM bound to PMCA-NPTN complexes in a Ca^{2+} -independent manner, and also observed fully active pumps with co-expressed CaM mutants lacking high-affinity Ca^{2+} -binding in all four EF-hands.~~

~~[1.10] The results in Figure 4 provide additional very nice independent support that the PMCA2 Ca²⁺ transport rates are similar to or greater than those for NCX based on similar quantities of each transporter.~~

~~Reply 1.10: This comment is well appreciated.~~

~~[1.11] Some additional explanation of why BK channels themselves don't distort cytosolic Ca signals may be helpful. Isn't this because the absolute number per cell of BK channels able to bind Ca vs. numbers of fluorescence reporters that are required to obtain a signal are vastly different?.~~

~~Reply 1.11: The reviewer's assumption is correct, a respective note was added to the revised ms.~~

~~[1.12] I would recommend the use of "extrude" or "remove" rather than "clear off intracellular Ca²⁺".~~

~~Reply 1.12: The wording was changed.~~

~~[1.13] Terming BK channels "native Ca sensors" seems to imply they are native to the COS cells.~~

~~Reply 1.13: We meant to emphasize the difference between BK_{Ca} channels and the many types of soluble 'dyes' usually applied for reading out Ca²⁺ concentrations (with all concomitant shortcomings).~~

Referee #2

~~While they seem to have conducted the functional assays and EM analysis soundly, I have a major concern about the novelty of this work. Indeed, the same methodology as that shown in Fig. 1b was previously reported by the same group (Schmidt N. et al. Immunity 96, 827-838 (2017)). They here developed another assay by employing a membrane-tethered Lek-GCaMP6 in place of the BK_{Ca} channels for Ca²⁺ sensing. However, few essential new insights seem to be gained into the kinetics and mechanisms of PMCA-mediated Ca²⁺ clearance by this alternate method. Moreover, although their computational modeling apparently explains the kHz-range Ca²⁺ extrusion by the PMCA-NPTN pump, they employed several assumed parameters that are not fully verified experimentally. Additionally, they used machine-learning procedures to calibrate the transition rates in a model of the BK_{Ca} gating. Given the lack of novelty and new mechanistic insights and the somewhat arbitrary computational modeling of the PMCA-mediated Ca²⁺ transport cycle, I cannot be positive about the publication of the present version of manuscript for publication in this journal. My additional major and minor concerns are listed below.~~

~~**Reply to general comment:** The statement of ‘*lacking novelty*’ based on our work in *Neuron* (not *Immunity*!) is nothing but incorrect – as this work introduced Neuroplastin/Basigin as essential subunits of native PMCA, but had to leave open whether transport efficiency is based on effective trafficking (of the complexes to the plasma membrane) or transport speed.~~

~~In fact, Ca^{2+} transport in the milli-second range (shown here for the first time!), together with accurate determination of the required number/density of PMCA are the major advances of this work. The reviewer’s statement of ‘*few essential new insights into the kinetics*’ is therefore hardly acceptable (or deliberately misleading), as is the judgement of parameters used for computer modelling *not being ‘fully verified’* (all derived from the data explicitly shown in figures).~~

~~[2.1] In the immunogold EM image displayed in the lower panel of Fig. 2a, it is hard to distinguish the particles of PMCA and Cav2.2 in CHO cells. Hence, it is hard to accept the authors’ statement that Ca^{2+} -source and Ca^{2+} -transporter are in nearly equal amount in their testing system.~~

~~**Reply 2.1:** In immediate contrast to the reviewer’s statements/assumptions, the distinct gold particles labelling the voltage-gated Cav2 channels (\varnothing 6 nm) and PMCA (\varnothing 12 nm) are readily distinguished and used for protein counting in EM – as is absolute state-of-the-art in the field (eg. *iScience* 22, 256; *Microscopy* 71, 172; *Neuron* 112, 755; *Neuron* 104, 680; *Neuron* 111, 2544; *Front Cell Neurosci* 9, 315). To further emphasize this distinction, the inset of Fig. 2a was increased in the revised ms.~~

~~[2.2] No error bars are shown in the bar graphs of Fig. 2b and 2c. Was this quantitative mass analysis performed only once? Statistical analysis is required. In this context, a list of identified peptides from PMCA1-4, NCX1-3, and SERCA1-3 and the abundance of each of the peptides need to be shown additionally.~~

~~**Reply 2.2:** The quantitative MS data (as detailed in Methods) were acquired from pooled mouse brains in duplicate measurements; data for each protein are based on more than 100 peptides (!) leading to highly accurate and reliable quantification (deviation between individual measurements less than 5%). The individual data points obtained for PMCA1-4, NCX1,2 and SERCA2,3 were added to the revised Figure 2. All MS data were deposited to the PRIDE database prior to submission and are available to the public upon acceptance.~~

~~[2.3] In computational modeling shown in Fig. 3, some parameters such as the cell volume and number of incoming Ca^{2+} are likely essential for calculation. Hence, the validity of these assumed or cited parameters needs to be discussed carefully in the main text.~~

~~**Reply 2.3:** In contrast to the reviewer’s statement, all parameters for computer modelling are given in detail in the Suppl. Tables 1-3, and the numbers of Ca^{2+} ions are explicitly discussed (in the Discussion section).~~

~~Minor points:~~

~~(4) The label of the scale bar in Fig.1c was explicitly stated in the legend; it was moved to the figure in the revised ms.~~

~~(5) The statistical statement was part of the 'reporting summary'; a respective note was added to the Methods section of the revised ms.~~

~~(6) The URL was updated.~~

Referee #3

~~Early estimates of PMCA turnover rates, mostly in red cell membranes or intact red cells, rendered values between 100 and a few thousands per second. Against such a background of uncertainty, it remained an open question whether the PMCA surface density and turnover rates of PMCA in neurons could account for the observed clearance rates of $[Ca^{2+}]_i$ after signalling events. To answer these questions Constantin et al., measured the turnover rate of the PMCA using a heterologous expression setup in CHO cells, and also the surface density of pumps in neurons.~~

~~The experiments were elegantly designed, impeccably executed, analysed, interpreted, reported and discussed in detail. The writing is clear, a pleasure to read. The answers on PMCA surface densities, and turnover numbers in the order of 5 to 20 thousand/s amply accounted for the measured $[Ca^{2+}]_i$ restoration rates post-signalling.~~

~~[3.1] I only have a minor comment concerning the operation of the PMCA as an electroneutral $Ca^{2+}:2H^+$ exchanger, also shown for neurons (J Physiol 587.2 (2009) pp 315-327). This issue is bypassed in the paper. It seems perfectly acceptable to ignore the H^+ displacements from EGTA on calcium binding in some of the protocols, but can transient effects of sharp H^+ accumulations at the inner membrane surface during peak pumping rates also be ignored?~~

~~Reply 3.1: Intracellular acidification resulting from the $Ca^{2+}:2H^+$ counter-transport by PMCA is counteracted by 5 mM HEPES in the pipette solution effectively buffering the pH in the physiological range. Potential effects of local pH changes are beyond the resolution of our experiments.~~

Point-to-point response to the reviewers' comments

We thank the two reviewers for the positive reception of the revised manuscript. Here, we address their final concerns.

Referee #2

After careful reading of the revised manuscript and rebuttal letter, I found that while the authors responded to some of my critical comments appropriately, the paper seems to require further revisions before recommendation for publication in this journal as described below.

[2.1] Regarding Reply 2.1, I do not agree at all with the authors' statement that "in the case of complex strictures such as PMCA, it is difficult to show densities and models superimposed plus side chain labels without losing clarity". There should be some ways to clearly highlight local structures with density maps and models superimposed even when molecular size of a target protein is quite huge, as actually done by the authors in supplemental Fig. 10. In this regard, revised Fig. 3c & d, Fig. 4c and Fig. 5d are still unacceptable. In particular, the positions and orientations of the Q351, N841, Q837 and D873 side chains seem critical to discuss the mode of interaction between PIP2 and PMCA2 and the PIP2-mediated activation of Ca²⁺ transport by PMCA2. Similarly, to discuss the counterproton binding sites and water-lined release pathway in PMCA2, the density needs to be shown clearly for the side chains of Q835, S877 and R969 as well as water molecules in Fig. 4c. Owing to the lack of well-resolved density maps for the residues above, the authors did not address my comments 2.1, 2.8, 2.9, and 2.10 in a well convincing manner.

Reply 2.1: We agree with the reviewer that showing protein models overlaid with cryo-EM density maps is generally insightful for the expert reader. We have therefore updated some of the figures mentioned by the reviewer (Fig. 3c & d, Fig. 4c) to include map-to-model overlays, also for the relevant amino acids.

The reviewer also requests a map-to-model overlay for Fig. 5d to highlight the orientations of Q351, N841, Q837 and D873. While we have attempted this for Fig. 5d, it simply did not yield an easy-to-understand figure. Therefore, in the previous revision seen by the reviewer, we had already included Supplementary Fig. 15a (Extended Data Fig. 7a in the new version), which shows a clipped map-to-model overlay for these important residues. The legend of Fig. 5d also refers directly to this Extended Data Figure. Therefore, expert readers can directly assess the quality of the structural data, while the main figure remains easy-to-understand for the broad readership of *Nature*.

With these final changes, we believe that the previous comments by the reviewer are all convincingly addressed.

[2.2] Regarding Reply 2.5, it is still hard to understand why the pK_a value of D873 largely changes from 3.8 to 7.0, which seems inconsistent with the values displayed in supplementary Table 6 (6.07 in E1 to 6.9 in E2P).

Reply 2.2: We thank this reviewer for spotting this typo. This was a remainder from an initial version of the manuscript where we used the (standard) pKa value of free aspartic acid for comparison. The calculated values in Supplementary Table 6 (now Supplementary Table 4) are correct and we corrected the typo in the manuscript. To further support the importance of the residue D873 as well as Q837, we prepared single replacement mutants (D873K and Q837A) and showed that PMCA2-mediated Ca²⁺-transport is severely impaired by these mutations (Figure 5e).

[2.3] Page 8, line 207: The absence of site I in SPCA1 is first reported by Chen et al. Sci. Adv. (2023) eadd9742. This paper should be cited.

Reply 2.3: Thank you for pointing this out. We have included the reference.

Referee #3

Overall, the manuscript is greatly improved in its revised form. Many of the issues with the structures and their presentations have been corrected. If a few minor issues (listed below) can be corrected, the manuscript would be suitable for publication.

[3.1] Figure 2c remains hard to interpret. The four domains move in distinct ways and it is not clear from the figure which domain is moving in which way. Perhaps a vector diagram depicting the per-residue or per secondary structural element movements may aid the reader in visualizing this ensemble of complex movements. Also, the transmembrane domain of the E1-Ca-ATP state is grey in panel d. It would be helpful to keep the coloring consistent.

Reply 3.1: This is a great suggestion. We have now added a separate panel in Fig. 2c in which we show C-alpha atom movements between the two states using arrows. We believe that this nicely captures the complex movements of PMCA2, also in combination with the Supplementary Videos.

We understand that the grey color in panel d may be confusing, especially because NPTN is also colored grey in other figures. We are required to use different colors here to clearly distinguish the two states. We have therefore changed the color from grey to purple.

[3.2] In Figure 3b, the dashed cyan line is described as a hydrogen bond, but it is not clear which residue, along with E348, participates in the interaction. In Figure 2c, it is not clear if the side chain or backbone of S936 interacts with R244. These details should be clarified.

Reply 3.2: We apologize for this oversight in Fig. 3b. E348 forms a hydrogen bond with the amine moiety of G1028. Glycine residues are notoriously difficult to depict when secondary structure elements are shown. We have therefore hidden the cartoon alpha helix for G1028 to highlight this interaction.

In Fig. 2c, the backbone of S936 interacts with R244 of NPTN. This is now specified in the figure.

[3.3] In Supplementary Figure 14, the PIP₂ molecule is oriented oppositely from in the panels c-e. Please make the orientation consistent.

Reply 3.3: We now show PIP₂ with the phosphate groups oriented up.